# Gut microbiota mediates intermittent-fasting alleviation of diabetes-induced cognitive impairment

Zhigang Liu [1,7,8✉], Xiaoshuang Dai [2,7], Hongbo Zhang[1], Renjie Shi[1], Yan Hui [2,3], Xin Jin[2], Wentong Zhang [1], Luanfeng Wang [1], Qianxu Wang[1], Danna Wang [1], Jia Wang[1], Xintong Tan [1], Bo Ren [1], Xiaoning Liu [1], Tong Zhao [1], Jiamin Wang[1], Junru Pan [1], Tian Yuan[1], Chuanqi Chu[1], Lei Lan[2], Fei Yin [4], Enrique Cadenas[5], Lin Shi [6,8✉], Shancen Zhao[2,8✉] & Xuebo Liu [1,8✉]

Cognitive decline is one of the complications of type 2 diabetes (T2D). Intermittent fasting (IF) is a promising dietary intervention for alleviating T2D symptoms, but its protective effect on diabetes-driven cognitive dysfunction remains elusive. Here, we find that a 28-day IF regimen for diabetic mice improves behavioral impairment via a microbiota-metabolites-brain axis: IF enhances mitochondrial biogenesis and energy metabolism gene expression in hippocampus, re-structures the gut microbiota, and improves microbial metabolites that are related to cognitive function. Moreover, strong connections are observed between IF affected genes, microbiota and metabolites, as assessed by integrative modelling. Removing gut microbiota with antibiotics partly abolishes the neuroprotective effects of IF. Administration of 3-indolepropionic acid, serotonin, short chain fatty acids or tauroursodeoxycholic acid shows a similar effect to IF in terms of improving cognitive function. Together, our study purports the microbiota-metabolites-brain axis as a mechanism that can enable therapeutic strategies against metabolism-implicated cognitive pathophysiologies.

---

[1] College of Food Science and Engineering, Northwest A&F University, Yangling 712100, China. [2] BGI Institute of Applied Agriculture, BGI-Shenzhen, Shenzhen 518120, China. [3] Department of Food Science, University of Copenhagen, Copenhagen 1958, Denmark. [4] Center for Innovation in Brain Science and Department of Pharmacology, University of Arizona, Tucson 85721 AZ, USA. [5] Pharmacology and Pharmaceutical Sciences, School of Pharmacy, University of Southern California, Los Angeles 90089 CA, USA. [6] Division Food and Nutrition Science, Department of Biology and Biological Engineering, Chalmers University of Technology, Gothenburg SE-412 96, Sweden. [7] These authors contributed equally: Zhigang Liu, Xiaoshuang Dai. [8] These authors jointly supervised this work: Zhigang Liu, Lin Shi, Shancen Zhao, Xuebo Liu. ✉email: zhigangliu@nwsuaf.edu.cn; shlin@chalmers.se; zhaoshancen@genomics.cn; xueboliu@nwsuaf.edu.cn

The prevalence of type 2 diabetes (T2D) is on the rise worldwide[1], and cognitive decline is a severe complication of T2D[2,3]. The etiology of cognitive impairments resulting from diabetes is multifactorial, including brain insulin resistance, cerebral microvascular damage, and neuroinflammation[2,4]. Previous research has found that T2D-associated or high-fat-diet-induced cognitive deficits are connected to brain insulin resistance, glucose uptake decrease, neurotransmitter metabolism disturbance, and mitochondrial dysfunction[5,6].

Recent studies have emphasized the critical impact of gut microbiota on regulating systemic insulin sensitivity and energy metabolism via the generation of functional microbial metabolites in diabetic rodent models and humans[7,8]. Moreover, gut microbiota plays a vital role in linking diet and host physiology and pathology. Microbiota homeostasis is essential for the maintenance of gut health and for modulating cognitive functions via the regulation of the permeability of the blood–brain barrier, brain energy homeostasis, and synaptic transmission[9–11]. It is noteworthy that interindividual variation of microbiota composition and microbial metabolites, such as short-chain fatty acids (SCFAs), 3-indolepropionic acid (IPA), secondary bile acids (BAs), and serotonin (5-HT), are critically driven by the genotype, age, and dietary patterns of the host[10,12]. They are also strongly linked to whole body metabolism and function particularly, and have been shown to have potential neuroprotective effects. BAs such as tauroursodeoxycholic acid (TUDCA) has been suggested as a potential agent in preventing amyotrophic lateral sclerosis and Alzheimer's diseases (AD)[13,14]. In addition, previous research demonstrated that IPA, one of the microbial deamination metabolites of tryptophan, might possess potential neuroprotective against β-amyloid-induced neuronal damage via scavenging free radicals[15].

Dietary restrictions have been well reported to be beneficial in the context of metabolic syndrome and the lengthening of lifespans, and gut microbiota may play a pivotal role in this[16]. Intermittent fasting (IF) regimens represent a periodic dietary restriction, which have shown to increase lifespan, promote energy metabolism, and to reduce the risk of developing various age-related pathologies[17–19]. It has been suggested that IF could ameliorate insulin resistance, activate adipose tissue browning, protect central nervous system autoimmunity, and improve diabetes-related retinopathy in rodent models, via modulation of the gut microbiome and microbial metabolites compositions[20–22]. Moreover, IF designed as alternate-day fasting has shown to improve age-related and Alzheimer's disease (AD)-associated neuroinflammation, brain structure, and cognitive functions in animal models[23–26]. Nonetheless, the protective effects of IF on the cognitive deficits associated with diabetes remain unclear, and to what extent the gut microbiota affect brain function is largely underexplored.

Given emerging reports that link gut microbiota to brain function and metabolism in relation to IF and health status, we undertook a comprehensive study to examine the impact of IF on diabetes-driven cognitive impairment and underlying mechanisms in db/db mice, a typical T2D rodent model, using a multi-OMICS analysis of the gut microbiota, plasma metabolome, and brain transcriptome combined with conventional behavioral tests. To determine the role of gut microbiota in mediating the protective effects of IF on cognitive deficiency, we also examined behavioral alterations in diabetic mice in which the gut microbiota was removed by antibiotics treatment. Our findings suggest that gut microbiota/metabolites mediate the neuroprotective actions of IF on diabetes, thus highlighting the potential of modulating the gut microbiota as an effective intervention for metabolism-implicated neurodegenerative diseases.

## Results

**IF suppressed insulin resistance and cognitive impairment.** Three-month-old db/db mice were fed either an ad libitum diet or fasted at 24 h intervals for 28 days (Fig. 1a). Compared with the db/m group, the littermate control mice, the food/energy intake was higher in db/db mice (Fig. 1c, d) (ANOVA with Tukey's test, $p < 0.01$). IF decreased the bodyweight of db/db mice (Fig. 1b) (ANOVA with Tukey's test, $p < 0.01$) without changing their total energy intake (Fig. 1c, d). In addition, IF treatment substantially reduced water intake per day in db/db mice (Supplementary Fig. 1A) (ANOVA with Tukey's test, $p < 0.01$) with less urine at the end of the treatment (Supplementary Fig. 1A).

The effects of IF on insulin resistance in the db/db mice were assessed with an insulin-tolerance test on day 28 (Fig. 1e). Specifically, IF treatment increased insulin sensitivity in db/db mice by suppressing fasting glucose levels (54.8% lower) and fasting insulin levels (29.8% lower), compared with ad libitum fed mice (Fig. 1f, g). The HOMA-IR values, which reflected insulin resistance, were lowered after IF treatment in db/db mice (Fig. 1h). Furthermore, IF significantly reduced the mass of epididymal white adipose tissues (eWAT) and the size of adipocytes (Supplementary Fig. 1B–D) (ANOVA with Tukey's test, $p < 0.01$). Morris water-maze tests were performed to assess the effects of IF on the cognitive impairment inherent in the diabetic mice. IF decreased escape latency in the 5-day navigation test, indicating that the improved cognitive deficits of db/db mice (Fig. 1i; Supplementary Fig. 1E). On the probe trial day, IF elevated the time spent in the target quadrants (ANOVA with Tukey's test, $p < 0.05$) in db/db mice, reflecting an increase in the spatial memory of diabetic mice (Fig. 1j). The results from the elevated plus maze, a behavioral test of anxiety, suggesting that IF treatment improved anxious behavioral and locomotor activity in db/db mice (Supplementary Fig. 1F–H).

**IF improved synapse ultrastructure and insulin signaling.** The ultrastructure of synapses in the hippocampus—the major region of the brain involved in cognition and memory—was examined, after the mice were killed. An analysis of postsynaptic density (PSD) (Fig. 2a) revealed that both the length and width of PSD were elevated in db/db-IF mice compared with the db/db group (Fig. 2b, c) (ANOVA with Tukey's test, $p < 0.01$).

Insulin signaling was assessed in the hippocampus in order to evaluate whether IF could ameliorate brain insulin resistance. IF increased the IRS1- Tyr$^{896}$/IRS1 and Akt phosphorylation, indicating that it stimulated the hippocampal insulin signaling pathway (Fig. 2d). We also observed that IF treatment enhanced the expression of BDNF, a neurotrophic factor involved in maintaining neuronal survival and synaptic function, and the phosphorylation of ERK/CREB, an upstream signaling of BDNF synthesis, compared with db/db mice (Fig. 2e) (ANOVA with Tukey's test, $p < 0.05$). Likewise, the expressions of PSD-95, a scaffolding protein in the excitatory postsynaptic density, was also increased in IF-treated mice hippocampi (Fig. 2e), in line with PSD ultrastructural alterations.

It is also worth noting that NFκB, a major transcriptional factor mediating inflammation[27], was activated in the hippocampus of diabetic mice (Supplementary Fig. 2). IF treatment suppressed NFκB activation and downregulated the phosphorylation of JNK/p38 and protein expressions of Iba-1, a well-known marker of microglial activation (Supplementary Fig. 2).

**IF promoted mitochondrial biogenesis in the hippocampus.** RNA sequencing on mice hippocampi was performed in order to identify the key biological processes and pathways regulated by IF in the hippocampus (Supplementary Data 1). After mapping

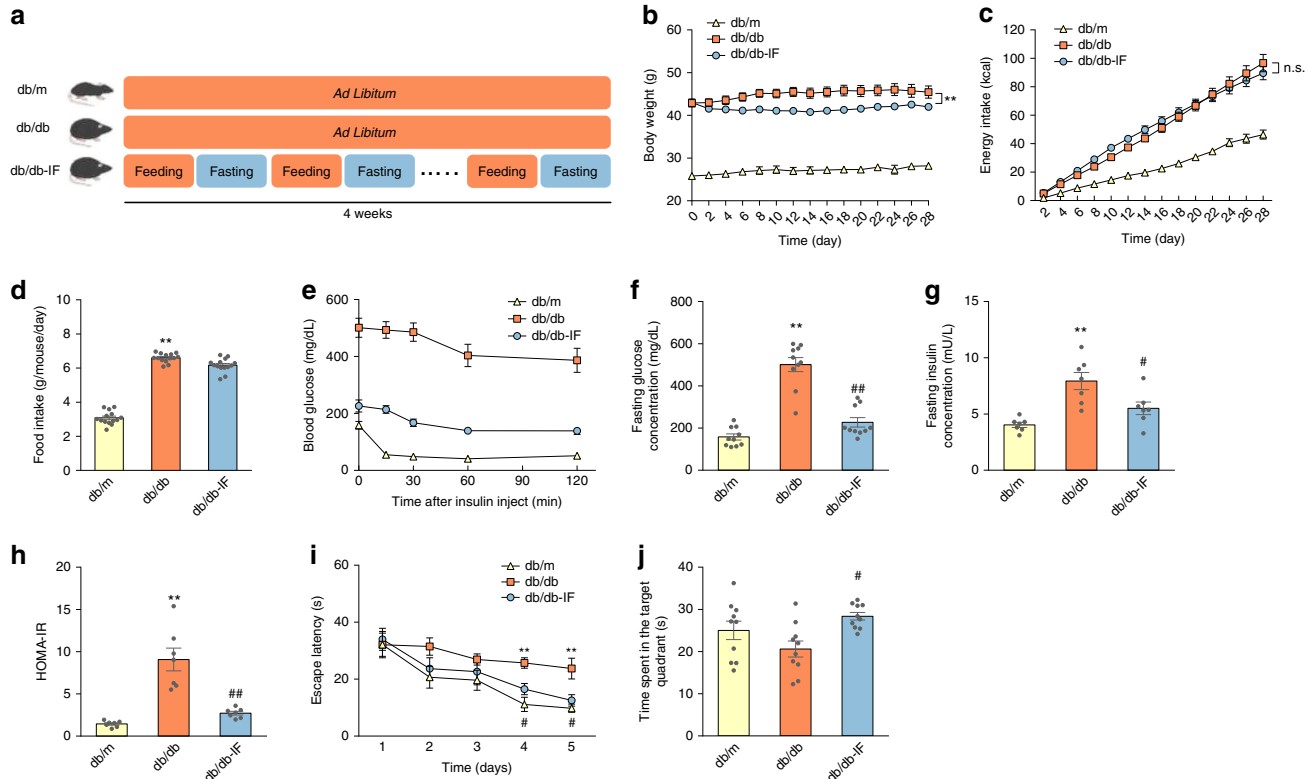

**Fig. 1 Intermittent fasting alleviated insulin resistance and cognitive impairment in db/db mice. a** Timeline depicting the diet of IF or ad libitum in each group; **b** bodyweight ($n = 10$ mice per group); **c** energy intake; **d** food intake; **e** insulin-tolerance test; **f** fasting glucose; **g** fasting insulin level ($n = 7$ mice per group); **h** HOMA-IR ($n = 7$ mice per group). The animals' cognitive functions were assessed by the Morris water-maze test as described in "Methods" ($n = 10$ mice per group); **i** escape latency (s), and **j** the time spent in the target quadrant (s) during the probe trial were recorded. Data presented as mean ± SEM. *$p < 0.05$, **$p < 0.01$, compared with db/m group, #$p < 0.05$, ##$p < 0.01$ compared with the db/db group. Significant differences between mean values were determined by one-way ANOVA with Tukey's multiple comparisons test. See also Supplementary Fig. 1. Source data are provided as a Source Data file.

310.85 Gb clean RNA-SEQ reads of all mice against the Mus musculus genome, we detected 27,094 genes (including 1345 new predicted genes with no annotation) with a FPKM value (Supplementary Data 2). Among them, 1181 genes were found to be highly expressed in db/db-IF mice compared with db/db and db/m mice, using differentially expressed gene (DEG) analysis (DEG-group 1 genes) (FDR-$p < 0.05$) (Fig. 3a; Supplementary Data 3), most of which were enriched in mitochondrial-related GO terms. The GO terms analysis of those genes that had lower expression in db/db mice (DEG-group 6 genes) indicated the respiratory chain and mitochondrial fission/translation biological process were dysregulated in db/db mice, which suggested abnormal mitochondrial metabolism in diabetic mice hippocampus. However, the related mitochondria-related genes expressions were corrected by the IF regimen (Fig. 3a). Moreover, the expressions of 483 genes in db/db mice hippocampus were comparable with those levels in db/m mice after IF treatment (DEG-group 3 and group 6) (FDR-$p < 0.05$), and 31.3% of these genes were even significantly higher (60 genes) or lower (91 genes) expressed than db/m mice (Supplementary Data 3). IF also elevated genes related to the KEGG pathway of oxidative phosphorylation (OXPHOS) via upregulating mitochondrial genes expression (Fig. 3b). The KEGG analysis of these genes were also indicated that IF treatment improved energy metabolism-related genes expressions in OXPHOS that were downregulated in db/db mice (Supplementary Fig. 3B).

To identify the biological networks affected by IF regimens, we performed a weighted gene co-expression network analysis (WGCNA) using all 27,094 genes in 31 samples, and 5 modules

were identified that contained functionally and biologically linked genes (Supplementary Fig. 3C). Of note, the "MEbrown" module was highly positively correlated with the IF regimen ($r = 0.892$, $p = 2e\text{-}11$) (Supplementary Fig. 3D), which consisted of 1044 genes (88.40%) identified using DEG (Supplementary Fig. 3E). We identified 49 hub genes (FDR-$p < 0.01$) with high inter-connectivity and intra-connectivity in the module (Supplementary Fig. 3E). Importantly, the majority of identified hub genes were enriched in the mitochondria and metabolism-related GO terms, and KEGG pathways of oxidative phosphorylation, Huntington's disease, AD, Parkinson's disease (Fig. 3c, d; Supplementary Data 5).

Consistent with the results from the RNA-sequencing analysis, the qPCR analysis confirmed that the mitochondrial and metabolic genes expressed were upregulated by IF in db/db mice (Supplementary Fig. 3F). As an indicator of mitochondrial biogenesis, the mtDNA/nDNA ratio in the hippocampus was increased by IF in db/db mice (Fig. 3g) (ANOVA with Tukey's test, $p < 0.01$), suggesting that the upregulated mitochondrial genes were likely a consequence of increased mitochondria mass.

We then sought to examine PGC1α and AMPK/mTOR protein expression to uncover the potential underlying mechanism of upregulating mitochondrial biogenesis in the db/db-IF group (Fig. 3e). IF led to a significant increase in the protein expression of PGC1α, the master regulator of mitochondrial biogenesis; this established a strong correlation with mitochondrial-related gene upregulation (Supplementary Fig. 3F). In line with previous studies, IF increased the activation of AMPK, a kinase mediating cellular energy metabolism sensitive to fasting status (Fig. 3e) and

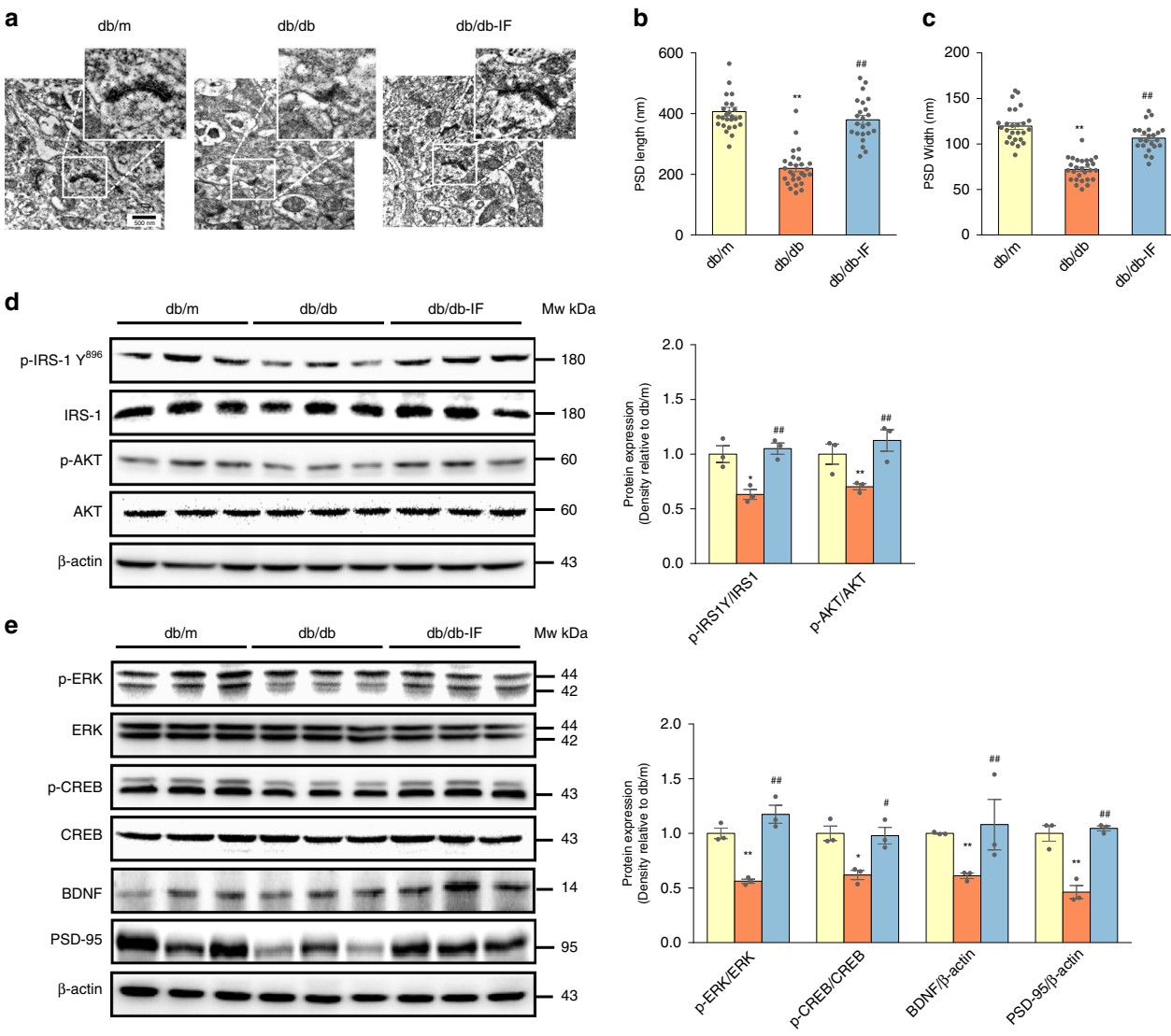

**Fig. 2 Intermittent fasting improved synapse ultrastructure and altered IRS/Akt and CREB/ERK signaling in db/db mice brain. a** Representative images of ultrastructure of synapse. **b**, **c** The length and width of PSD ($n = 6$ slices per group). **d**, **e** Western blots analysis of hippocampal IRS/Akt and CREB/ERK related signaling ($n = 3$ mice per group). Data presented as mean ± SEM. *$p < 0.05$, **$p < 0.01$, compared with db/m group, #$p < 0.05$, ##$p < 0.01$ compared with the db/db group. Significant differences between mean values were determined by one-way ANOVA with Tukey's multiple comparisons test. See also Supplementary Fig. 2. Source data are provided as a Source Data file.

inhibited the phosphorylation of mTOR, a eukaryotic cell nutrient sensor that plays an essential role in regulating mitochondrial quality control and OXPHOS process[28].

**IF-restructured gut microbiota and microbial metabolites**. The integrity of the gut barrier is critical to maintaining gut permeability and preventing endotoxemia, and is also related to diabetes and its complications[21]. It has also been reported that increased endotoxemia from increased gut permeability can increase inflammation, which triggers impaired glucose tolerance[29]. We thus investigated the effects of IF on gut barrier integrity, by assessing the villi length, the thickness of the muscularis, and the numbers of goblet cells. The results showed that IF increased the villi length and the muscularis thickness in diabetic mice, but had no influence on the numbers of goblet cell (Supplementary Fig. 4A–D). Moreover, the permeability of the colon was assessed with a use of Ussing Chamber, and we observed that IF prevented gut leakage accompanied with a decline in plasma LPS levels

(Supplementary Fig. 4G, H) (ANOVA with Tukey's test, $p < 0.01$). The expression of claudin-1, a tight-junction protein in gut barrier[30], was also elevated in db/db-IF colon tissue (Supplementary Fig. 4A, E, F), in line with the intestinal permeability alteration.

Gut microbiota composition was determined from fecal samples of the respective mice on day 0 (baseline level) and day 28 using bacterial 16S rRNA gene v3–v4 amplicon sequencing. Specifically, the microbiome alpha diversity significantly increased after a 28-day IF treatment even though the total number of OTUs remained at the same level initially (Fig. 4a) (Kruskal–Wallis, $p < 0.05$). The unweighted Unifrac distance of the db/db-IF mice on the day 28 was different from other groups, which indicates that the IF regimen changed β-diversity in the meantime (Fig. 4b) ($p < 0.05$). Considering time as a background effect, different diets and murine genotype were major factors in the shaping of gut microbiome and explained 5.9% and 7.7% of the total variance, respectively (Fig. 4c). IF treatment did not totally overturn the influence of diabetes with decreased relative abundance of *Allobaculum* and *Bifidobacterium* in both db/db

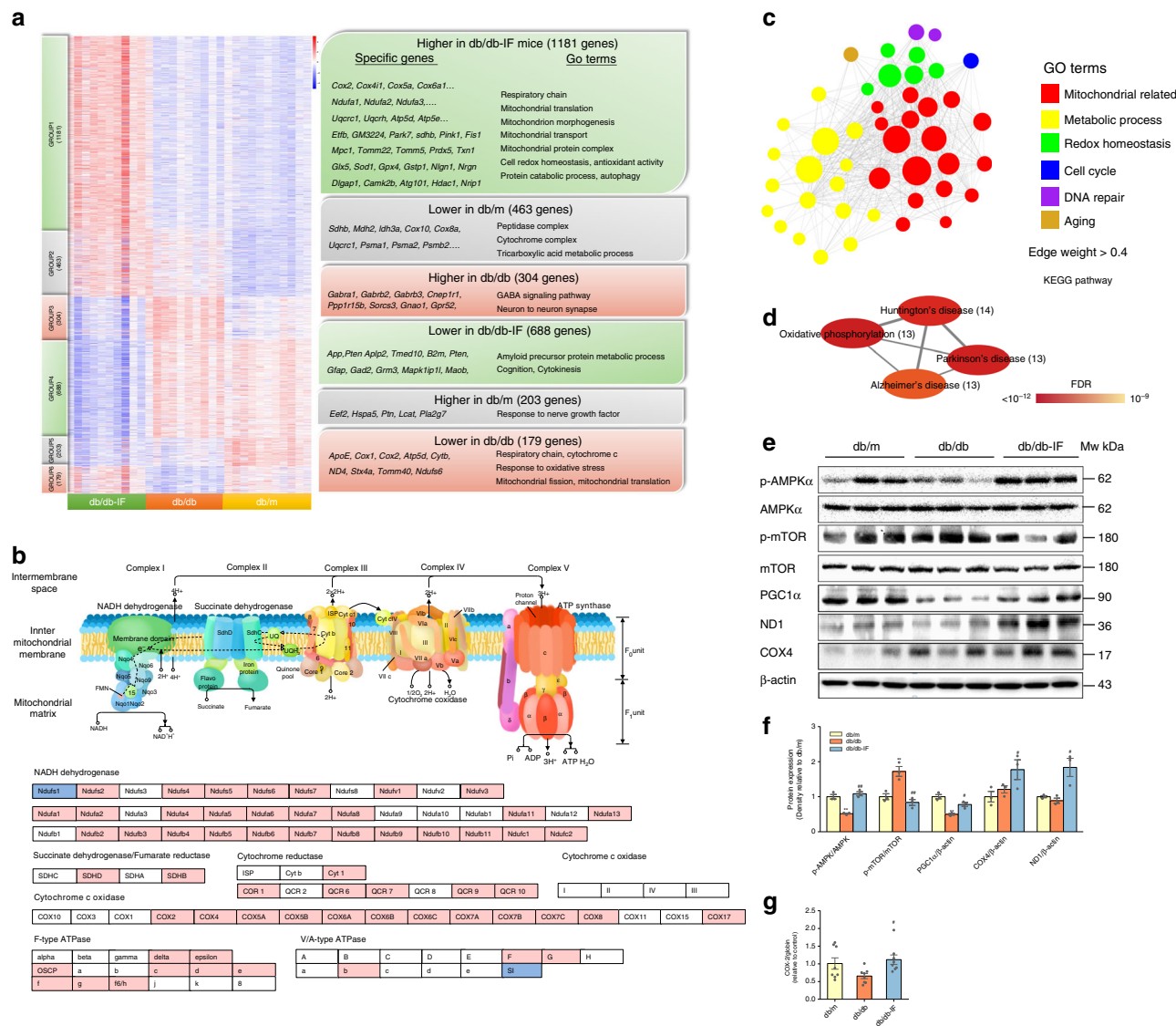

**Fig. 3 Intermittent fasting improved energy metabolism and mitochondrial biogenesis in the hippocampus. a** Heatmap displaying 2503 differentially expressed genes in the hippocampus among db/m, db/db, and db/db-IF groups ($n = 11, 9, 11$ mice in each group, respectively) that were clustered into six distinct gene groups through DEG analysis using Ballgown software (FDR-$p < 0.05$). Enriched GO terms (FDR-$p < 0.05$) and representative genes are shown on the right for each cluster. The $p$-values of GO terms were determined on the WebGestalt website; **b** KEGG of oxidative phosphorylation (FDR-$p < 0.05$) of group 1 (1181) and group 4 (688) genes based on DEG analysis. Pink represents upregulation during Intermittent fasting. Blue represents downregulation during Intermittent fasting. **c** Network and GO annotation of 49 brown-module hub genes (upregulated when intermittent fasting, r[ME and gene] > 0.8, r[trait and gene] > 0.85, FDR-$p < 0.01$) in WGCNA analysis (edge weights > 0.4). The $p$-values of GO terms were determined on the WebGestalt website; **d** KEGG analysis (FDR-$p < 0.05$) of the brown-module hub genes in the WGCNA analysis. The $p$-values of KEGG analysis were determined on the WebGestalt website; **e, f** Western blots of mTOR/AMPK/PGC1α signaling ($n = 3$ mice per group); **g** mitochondrial DNA levels in brain tissue ($n = 8$ mice per group). Data of **f, g** presented as mean ± SEM. *$p < 0.05$, **$p < 0.01$, compared with the db/m group, #$p < 0.05$, ##$p < 0.01$ compared with the db/db group. Significant differences in **d, g** between mean values were determined by one-way ANOVA with Tukey's multiple comparisons test. The boxplot elements are defined as following: center line, median; box limits, upper and lower quartiles; whiskers, 1.5 × interquartile range. See also Supplementary Fig. 3 and Supplementary Data 1–5. Source data are provided as a Source Data file.

mice (Fig. 4d). However, IF treatment improved *Lactobacillus* and butyrate-producing *Odoribacter* while decreased *Enterococcus*, *Streptococcus*, and unknown Enterococcaceae (Fig. 4d; Supplementary Data 6) (ANCOM, $p < 0.05$). Compared with the db/db group, bacteria from *Candidatus Arthromitus*, *Rummeliibacillus*, unknown Enterococcaceae, and Leuconostocaceae were lower in both db/m and db/db-IF groups (Fig. 4d).

Moreover, the identified IF-related genera were correlated with the obesity and cognition-associated blood glucose and body-weight. Specifically, *Candidatus Arthromitus* and an unknown

Leucostocaceae genera were positively correlated with blood glucose (Pearson's coefficient = 0.52 and 0.72, respectively, $p < 0.05$). An unknown Enterococcaceae genera was positively correlated with bodyweight (Pearson's coefficient = 0.59, $p < 0.05$) (Supplementary Fig. 5B, Supplementary Data 7).

Considering the mice gene-type effect, we evaluated the microbiota composition of the db/db mice and the db/db-IF group on day 28. We found that a total of 17 zOTUs affected by IF, 5 of which belonged to *Lactobacillus* (Fig. 4e). Meanwhile, a PICRUSt analysis revealed 11 differently abundant KEGG gene

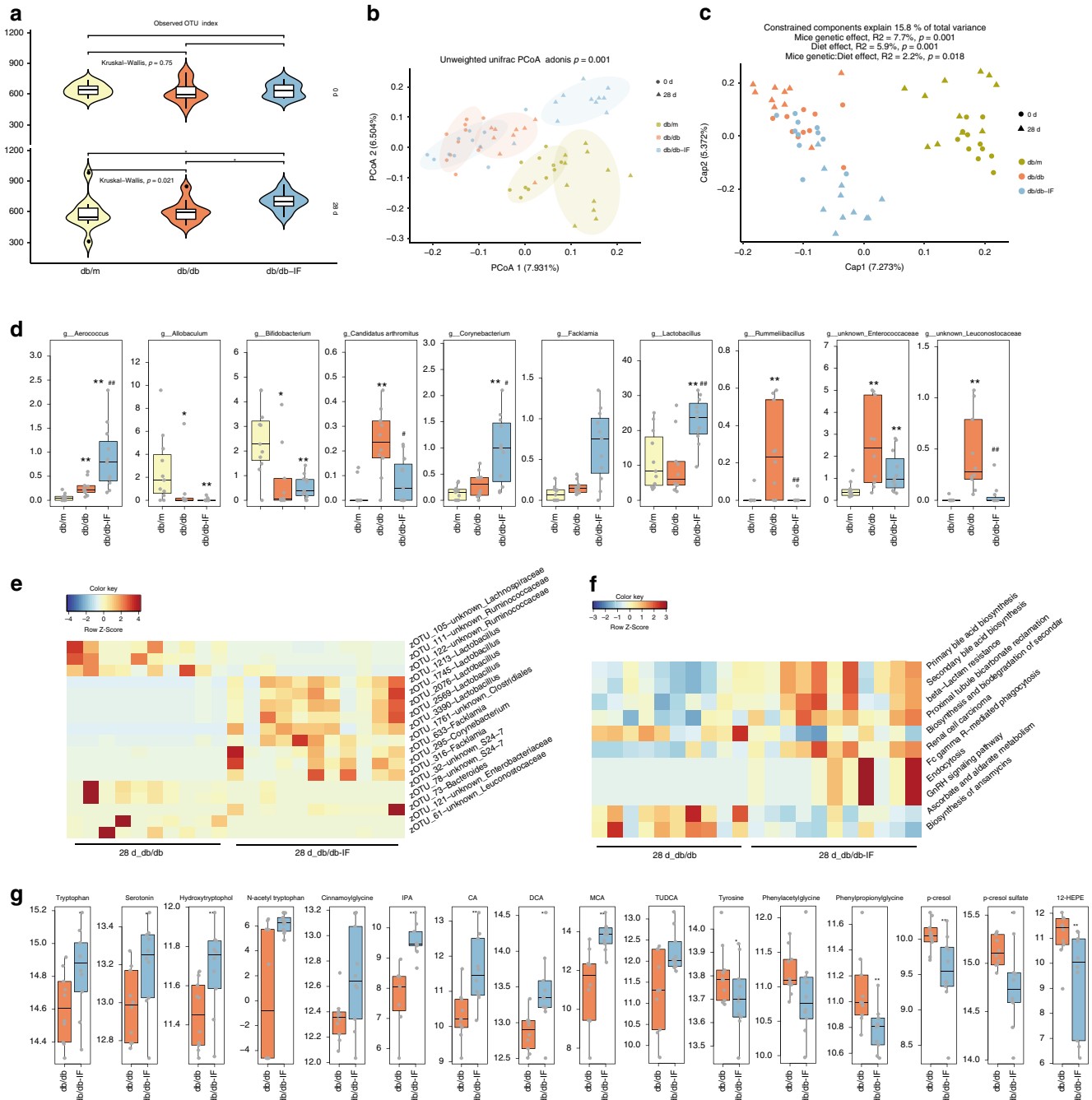

**Fig. 4 Gut microbiome and plasma metabolome analysis of intermittent fasting-treated db/db mice. a** Total number of observed OTUs in db/m, db/db, and db/db-IF ($n = 11, 10, 11$ mice in each group, respectively). Significant differences between day 0 and day 28 were tested by Kruskal-Wallis test. Significant differences between treatment groups were tested by Wilcoxon rank-sum test (*$p < 0.05$). **b** Principal coordinate analysis (PCoA) based on unweighted Unifrac distance and permutational manova (adonis) were used to test the difference in gut microbiota composition and diversity between groups (the colored ellipse denotes an 80% confidence within each group. **c** Constrained analysis of principal coordinate (CAP) of diet and mice gene type was performed on cumulative sum scaling (CSS) normalized Bray Curtis distance after the removing time effect. The permutational ANOVA-like test for constrained responcence analysis (ANOVA.CCA) was applied to examine the influence of diet and mice gene type on the matrix. **d** Relative abundance of different genera was identified by an analysis of composition of microbiome (ANCOM) on day 28 (FDR-$p < 0.05$). Wilcoxon rank-sum test was adopted for comparisons between groups (*$p < 0.05$, **$p < 0.01$, compared with the db/dm group, #$p < 0.05$, ##$p < 0.01$ compared with the db/db group). **e** A Z-score scaled heatmap of different zOTUs identified by ANCOM between db/db-IF and db/db on day 28 with FDR-$p < 0.05$. **f** A Z-score scaled heatmap of different pathways identified by an extraction of differential gene expression (Edge) between db/db-IF and db/db on day 28 with FDR-$p < 0.1$. **g** The difference in microbial metabolites in db/db and db/db-IF mice ($n = 10$ mice per group). Microbial metabolites that were up/downregulated after IF treatment. Differences were examined by Wilcoxon rank-sum test (*$p < 0.05$, **$p < 0.01$). IPA 3-indolepropionic acid, CA cholic acid, DCA deoxycholic acid, MCA muricholic acid, TUDCA tauroursodeoxycholic acid, 12-HEPE 12-hydroxyeicosapentaenoic acid. The boxplot elements are defined as following: center line, median; box limits, upper and lower quartiles; whiskers, 1.5 × interquartile range. See also Supplementary Figs. 4–6 and Supplementary Data 6–9. Source data are provided as a Source Data file.

pathways (FDR-$p < 0.1$). Among them, the primary and secondary bile acid biosynthesis were abundant in the db/db-IF group (Fig. 4f; Supplementary Data 8).

IF treatment demonstrated a great impact on gut microbiota, raising our interest in investigating whether and how, it could affect circulating microbial metabolites, whose levels were primarily modulated by the gut microbiota. This investigation could provide novel insights into the mechanism underlying the observed benefits of IF treatment on host metabolism. Overall, the untargeted metabolite profiling of plasma samples revealed marked changes in numerous metabolites after 28 days of IF treatment in db/db mice (Supplementary Fig. 5C, D). The number of metabolite features that differed between treatments were 540 and 491 out of 5604 and 5230 in a data set collected from the reverse-phase chromatography positive (RP +) and negative polarity (RP−), respectively (Supplementary Fig. 5E, F) (FDR-$p < 0.05$). Specific emphasis was placed on an a priori defined subset of 23 microbial metabolites whose levels were primarily modulated by the abundance of gut microbiota (Supplementary Data 9) to demonstrate a direct link between IF treatment, gut microbiota, and metabolome[12]. Among these metabolites, IF increased plasma levels of 5-HT, tryptophan, p-hydroxyphenyl acetic acid, N-acetyl tryptophan, cinnamoylglycine, IPA, and bile acids, e.g., cholic acid (CA), deoxycholic acid (DCA), muricholic acid (MCA), and TUDCA compared with ad libitum fed db/db mice (Wilcoxon rank-sum test, $p < 0.05$). In contrast, tyrosine, phenylacetylglycine, phenylpropionylglycine, p-cresol, and p-cresol sulfate, as well as 12-hydroxyeicosapentaenoic acid (12-HEPE) were lower in the db/db-IF group than the db/db group (Fig. 4g) (Wilcoxon rank-sum test, $p < 0.05$).

In addition, we also analyzed concentrations of short-chain fatty acids (SCFAs) in fecal samples, which have shown to protect gut barrier function and regulate insulin sensitivity[31,32]. After the 28-day treatment, IF improved levels of acetate, propionate, and butyrate in db/db mice (Supplementary Fig. 4I–K) (ANOVA with Tukey's test, $p < 0.05$). It should be noted that SCFAs concentrations were low in plasma, and thus could not be accurately detected using untargeted metabolomics in the current setting.

**Integrated multi-OMICs analysis on IF regimen**. Having identified different omics signatures that were upregulated or downregulated by IF in diabetic mice, the next step was to assess the interplay across gene expressions in the hippocampus (Fig. 3), gut microbiota compositions (Fig. 4), and microbial metabolites (Fig. 4), in relation to the observed impact of IF treatment on cognitive deficiencies in mice (Fig. 2). The multi-OMICs data set was integrated using the data integration analysis for biomarker discovery using a latent component method for OMICs (DIABLO). DIABLO is a multivariate dimension reduction discriminant analysis method designed to identify biologically relevant and highly correlated signatures from various OMICs techniques[33] (Fig. 5; Supplementary Information).

Prior to multi-OMICS integration, we first evaluated whether the IF-upregulated hub genes $n = 36$), the IF-altered OTUs ($n = 17$), the predefined plasma microbial metabolites ($n = 23$), and three fecal SCFAs, i.e., acetate, butyrate, and propionate could predict the IF status of mice in a multivariate manner[12]. Predictive modeling was conducted using a partial least square-discriminant analysis incorporated into a repeated double cross-validation framework (rdCV-PLSDA), which effectively minimized the risk of statistical overfitting[34]. A significant separation between db/db and db/db-IF mice was achieved with a predictive accuracy of 100%, 90%, and 95% for hub genes, OTUs, and metabolites, respectively (Fig. 5a). The optimal selected number of latent components was 1 for each of the OMICs data sets. The predictive ability of the constructed models outperformed those

of 1000 permuted models, demonstrating the robustness and validity of predictive models in discriminating db/db from db/db-IF using parsimonious sets of OMICs signatures with great generalizability (Student's $t$ test, $p < 0.05$, Supplementary Fig. 6B).

The integrative modeling was then performed on the multi-OMICs signatures, and one latent component comprised ten key predictors from each of OMICs data sets was identified, contributing a great separation between db/db and db/db-IF (Fig. 5b). The optimally selected key predictors included several species of gut microbiota, i.e., *Lactobacillus*, *Bacteroides*, and *Facklamia*, microbial cometabolites, i.e., bile acids, IPA, 5-HT, tryptophan, hydroxytryptophol, and indoxyl sulfate, and genes enriched in mitochondrial, ribosomal or metabolic-related GO terms (Fig. 5b, c; Supplementary Fig. 6C). Key predictors were also highly correlated (Fig. 5d).

**Gut microbiota are required for the beneficial effects of IF**. To demonstrate the crucial role of gut microbiota in mediating inherent diabetic cognitive deficits, we sought to investigate whether the observed beneficial effects of IF could be affected after removing the gut microbiota in db/db mice.

An antibiotics cocktail treatment was applied to establish the role of the microbiota in the neuroprotection afforded by IF[35]. The mice were administrated with antibiotics in the drinking water starting 14 days before the 4-week IF regimen and throughout the experiment, and the behavioral changes were assessed after 28 days (Supplementary Fig. 8A, B). The antibiotics treatment weakened the weight loss effects of the IF treatment (Fig. 6a) (ANOVA with Tukey's test, $p < 0.05$). The antibiotics treatment had no impact on the eWAT weight, but enhanced the liver and cecum weight (Supplementary Fig. 8E–G). Both IF and antibiotics treatment reduced food and water intake in db/db mice (Supplementary Fig. 8C, D). However, the administration of antibiotics had no impact on the beneficial effects of IF on insulin resistance (Supplementary Fig. 8H–K). Of note, the cognition improvement assessed using the Morris water-maze test associated with IF was partly abolished by antibiotics treatment, with longer escape times on the 5th day of the test and lower times in the target quadrant on the probe test day (Supplementary Fig. 6B; Fig. 8L, M). Moreover, mitochondrial biogenesis—upregulated in the db/db-IF mice—was abrogated with antibiotics treatment (Fig. 6c). An analysis of PSD (Fig. 6d, e) revealed that the width of PSD—elevated in db/db-IF mice—was reduced by antibiotics treatment (ANOVA with Tukey's test, $p < 0.01$). These results together indicate that the removal of microbiota partly abolishes the protective effects of IF on cognitive function.

In addition, compared with the db/db-IF group, we found that the removal of gut microbiota by the antibiotics treatment significantly reduced plasma IPA and fecal SCFAs in db/db mice (Supplementary Fig. 9, Supplementary Data 10) (ANOVA with Tukey's test, $p < 0.01$).

**Microbial metabolites improved the cognitive function**. An additional animal study was conducted to investigate the role of IF-regulated microbial metabolites, in particular, IPA and SCFAs that were reduced in the antibiotics-treated group, in improving cognitive deficits. Besides, the effects of peripheral 5-HT and TUDCA on cognitive function were also examined, since these metabolites were elevated in the IF group in this study (Fig. 4g). Previous studies have also reported the potential of their neuroprotective effects[13,14,36]. Administration of all these metabolites individually improved cognitive function and insulin sensitivity in db/db mice (Fig. 6g; Supplementary Fig. 10C–H) (ANOVA with Tukey's test, $p < 0.05$). Consistent with the beneficial effects of IF treatment, administration of these metabolites also enhanced mitochondrial

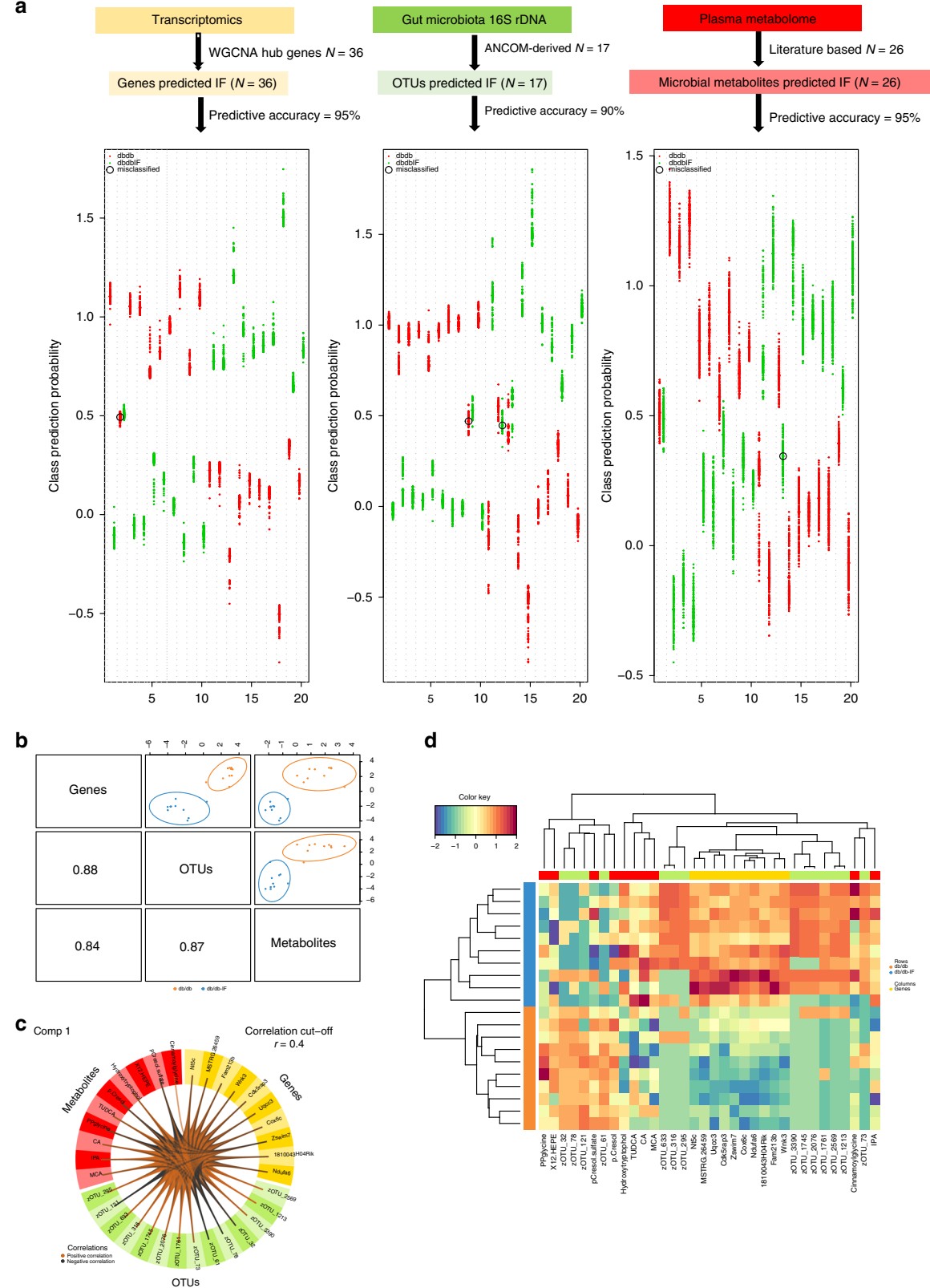

biogenesis and protected the ultrastructure of synapses (Fig. 6h–j). Moreover, treatment of TUDCA but not other tested metabolites suppressed the bodyweight gain in db/db mice (Fig. 6f). Besides, all the metabolites treatment had no influence on food intake and water intake of the db/db mice (Supplementary Fig. 10A, B). Although the dosage of the metabolites treatment was higher than that in IF regimen group mice, these results partly evaluate the causation and the roles of these metabolites in during IF regimen.

**Fig. 5 Strategy and model performance of the integrative modeling on multi-OMICS in relation to IF treatment. a** The performance of predictive models for three OMICs signatures on IF status. OMICs signatures include: 36 IF-upregulated hub genes from the IF-related brown module that was identified using WGCNA; 17 OTUs that were found to be influenced by IF treatment; 26 priori defined sets of microbial metabolites, including 23 plasma metabolites and 3 SCFAs measured in fecal samples. For each OMICs data set, multivariate predictive modeling was conducted using partial least square-discriminant analysis incorporated into a repeated double cross-validation framework (rdcv-PLS). Prediction performance is shown in downstream figures: each swim lane represents one mice sample. For each sample, class probabilities were computed from 200 double cross-validations. Class probabilities are color coded by class and presented per repetition (smaller dots) and averaged over all repetitions (larger dots). Misclassified samples are circled. Predictive accuracy was calculated as a number of correctly predicted samples/total number of measured samples. **b** The model performance of DIABLO integrative modeling on OMICs signatures in relation to IF. The use of DIABLO maximized the correlated information between multiple data sets, i.e., genes, OTUs and metabolites, while optimally identifying in a parsimonious set of key OMICs variables relevant for IF status, i.e., ten key predictors from each of OMICs data sets. Scatter plots depicting the clustering of groups, i.e., db/db and db/db-IF, based on the first component of each data set from the model showed a significant separation between groups. A scatterplot displays the first component in each data set (upper diagonal plot) and Pearson correlation between components (lower diagonal plot). **c** The Circos plot shows the positive (negative) correlation, denoted as brown (gray) lines, between selected multi-OMICs features. **d** A clustered image map (Euclidean distance, complete linkage) of the multi-OMICs signature. Samples are represented in rows, selected features on the first component in columns. See also Supplementary Fig. 7. Source data are provided as a Source Data file. Detailed procedure and R code are provided in the Supplementary Information.

## Discussion

A number of studies has shown beneficial effects of IF on diabetes and its complications through the restructuring of gut microbiota[20,21]. However, whether IF could alleviate diabetes-related cognitive deficits and the role played by gut microbiota is still unclear, highlighting an urgent need to investigate the mechanisms underlying how IF improves cognitive function. This study has demonstrated for the first time that a 28-day IF treatment alleviated diabetes-induced cognitive impairment via a microbiota–metabolites–brain axis, benefiting from comprehensive investigations on diabetic mice behavior/synaptic structure, mitochondrial/energy metabolism-related signaling, and an integrated analysis of multi-OMICs.

IF improved cognitive function and spatial memory in diabetic mice as evidenced from a Morris water-maze test, a de-facto standard and valid cognitive assessment[37,38]. Our previous study found that the impairment of brain insulin signaling led to glucose hypometabolism, disturbed tricarboxylic acid (TCA) cycle, as well as altered neurotransmitter levels and a decline in long-term potentiation (LTP)[39]. This finding provided an essential link between brain energy metabolism and cognitive function. Here, we found that IF stimulated IRS1/Akt signaling and its related ERK/CREB/BDNF neurotrophic signaling, and IF inhibited the NFκB/JNK inflammatory pathways (Fig. 2; Supplementary Fig. 2). Consistently, the increase of postsynaptic protein PSD-95 and the improvement of the ultrastructure of synapses in the hippocampus further demonstrated the benefits of an IF regimen on synaptic function. Further research is needed to determine the effects of IF on neuronal glucose uptake/metabolism and synaptic transmission in diabetic models via PET/CT/NMR and LTP assay measurements, respectively.

An integrated analysis of multi-OMICs data revealed strong links between the alterations of microbial metabolites in circulation and the IF-induced changes of the microbiome profile and hippocampus mitochondrial and energy metabolism-related gene expression. These findings suggest the beneficial effects of 28-day IF on diabetes-induced cognitive functions via the gut–metabolism–brain axis (Fig. 5), which to the best of our knowledge, has not been reported before.

Herein, we observed that a 28-day IF regimen improved gut barrier integrity and decreased plasma LPS level, which could partly explain how IF reduces neuroinflammatory responses. IF also altered microbiome diversity in diabetic mice, accompanied with altered bacteria abundance, enhancing the abundance of *Lactobacillus* (Fig. 4). The effect of IF on gut microbiota has been well reported[20–22]. However, the restructuring effect of IF on gut microbiota may be different, due to the variance in animal strains,

housing condition, baseline microbiota profiles, and feeding periods. Li et al. reported that an every-other-day fasting regimen increased the OTU abundance of *Firmicutes*, and at the same time decreased the majority of other phyla in the fecal contents of male C57BL/6N mice[20]. IF also enriched *Lactobacillaceae*, *Bacteroidaceae*, and *Prevotellaceae* families in the stool samples of female C57BL/6J fed every other day[22]. Intriguingly, a recent study of IF on db/db mice fecal microbiota composition found that a 7-month IF regimen led to the enrichment in *Lactobacillus* and the reduction of *Akkermansia*[21], which is consistent with our findings.

The IF-restructured gut microbiota led to alterations in the microbial metabolites in plasma, which might further explain how IF alleviates cognitive impairment in diabetic mice. The PICRUSt analysis found that the several microbes-related metabolism pathways including primary/secondary bile acid biosynthesis were altered after IF regimen (Fig. 4f). However, PICRUSt has its autologous defects on account of the availability of appropriate reference genomes and undetectable resolution of 16S rRNA sequences on strain variation. In this study, using metabolome analysis, we found that IF increased plasma levels of BAs including CA, TUDCA, MCA, and DCA, and these BAs were correlated with IF affected gut microbiome and transcriptome in the hippocampus. Moreover, administration of TUDCA also improved the cognitive impairment in db/db mice (Fig. 6). These findings suggest BAs play a role in microbiota–metabolites–brain axis in relation to cognitive impairment in diabetic mice. In line with our study, a long-term IF treatment prevented diabetic microvascular complications by improving plasma TUDCA[21]. The dietary-induced dysregulation of BA synthesis was found to be related to neuroplasticity impairment[40]. A recent human study has also reported a strong association between BA profiles and cognitive function, with the decreased plasma concentration of CA correlating particularly with AD progression[41]. In addition, BA homeostasis is essential for normal glucose/lipid homeostasis, which may partly explain why the energy metabolism-related genes were highly expressed in the db/db-IF group[42,43].

Other important microbial metabolites altered by IF treatment in db/db mice include tryptophan and its metabolites 5-HT and IPA. Tryptophan is not only an essential amino acid for protein synthesis but also is the precursor for a number of neuroprotective metabolites[10,44]. Gut microbiota plays a critical role in regulating the synthesis of 5-HT, a well-known key hormone/neurotransmitter of the gut–brain axis in regulating mood and cognition[45]. Interestingly, we found that IF increased 5-HT levels and suppressed anxiety in db/db mice (Fig. 1f–h; Supplementary Fig. 4G). We found that IF regimen enriched *Lactobacillus*, 5-HT

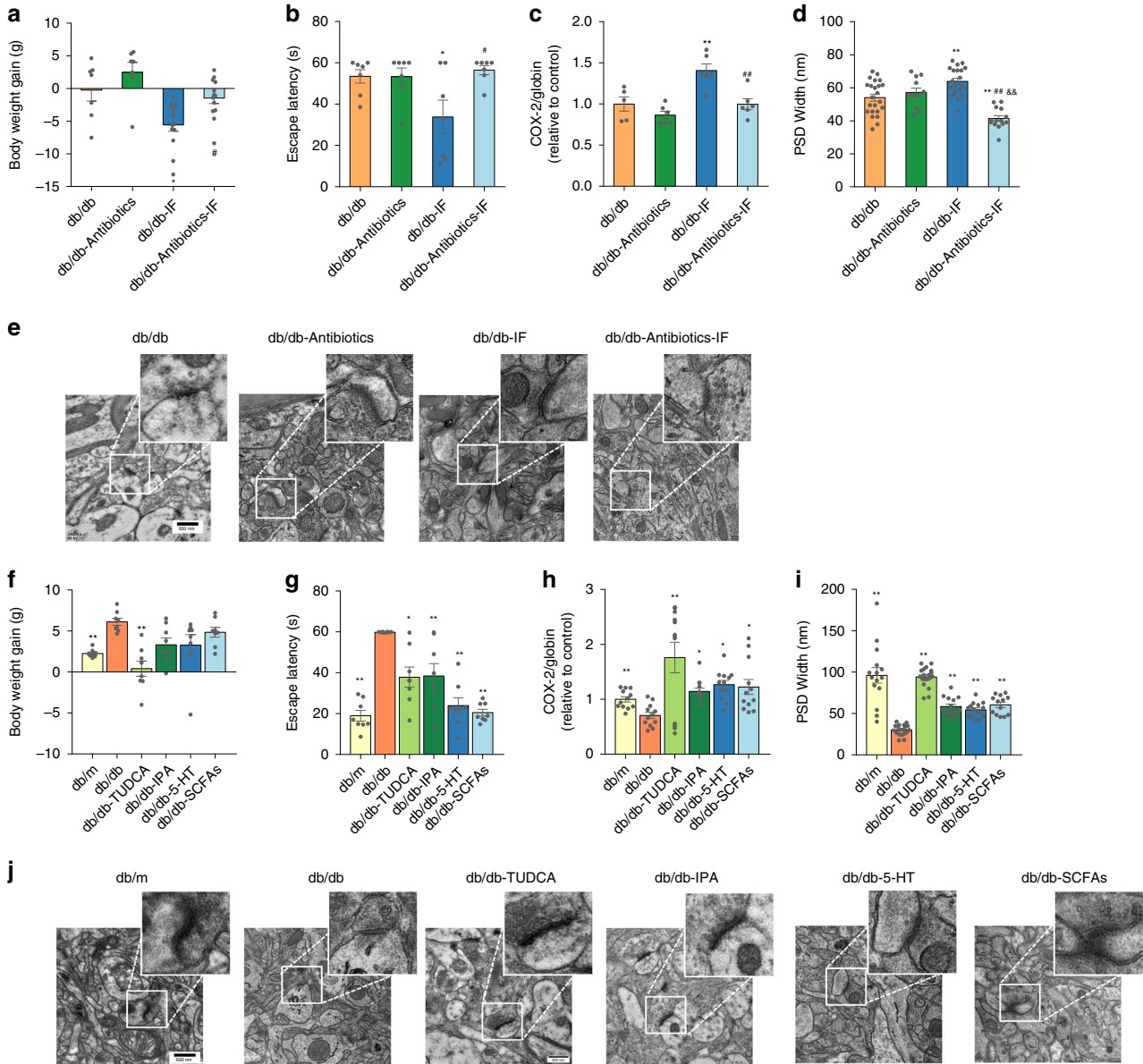

**Fig. 6 The effects of antibiotics and microbial metabolites on the cognitive functions of IF-treated db/db mice.** The mice were administrated with antibiotics in the drinking water starting 14 days before the 4-week IF regimen and throughout the experiment (the detailed antibiotics treatment is as described in "Methods"). **a** Bodyweight gain (g) ($n = 7$ mice and 13 mice in ad libitum feeding and IF regimen groups, respectively). **b** Escape latency (s) in place navigation test (the 5th day). **c** Mitochondrial DNA levels in brain tissue ($n = 7$ mice per group); **d** width of PSD. **e** Representative images of ultrastructure of synapse ($n = 6$ slices per group). Data presented as mean ± SEM. *$p < 0.05$, **$p < 0.01$, compared with the db/db group, $^&p < 0.05$, $^{&&}p < 0.01$, compared with the db/db- antibiotics group, $^#p < 0.05$, $^{##}p < 0.01$ compared with the db/db-IF group. Significant differences between mean values were determined by two-way ANOVA (IF regimen and antibiotics treatment as two factors) with Tukey's multiple comparisons test. The db/db mice were administrated with IPA, 5-HT, TUDCA, or SCFAs, i.e., acetate, butyrate, and propionate, individually ($n = 8$ mice per group). **f** Bodyweight gain; **g** escape latency (s) in place navigation test (the 5th day) ($n = 8$ mice per group); **h** mitochondrial DNA levels in brain tissue ($n = 8$ mice per group); **i** width of PSD; and **j** representative images of ultrastructure of synapse ($n = 6$ slices per group). Data presented as mean ± SEM. *$p < 0.05$, **$p < 0.01$ versus the db/db group. Significant differences between mean values were determined by one-way ANOVA with Tukey's multiple comparisons test. See also Supplementary Figs. 8–11, Supplementary Data 10. Source data are provided as a Source Data file.

and hippocampus gene expressions, supporting the previously reported effects of *Lactobacillus* genera on peripheral 5-HT synthesis and alleviating behavior impairments[46] (Fig. 5). Moreover, IPA, a microbial deamination metabolite of tryptophan, is a potent neuroprotective antioxidant. Higher plasma concentration of IPA was associated with lower type 2 diabetes risk in a recent study[44]. It has also been well reported for its role in preventing neuronal damage[15]. Consistently, we also found that IPA and 5-HT treatment significantly attenuated cognitive deficits in diabetic

mice (Fig. 6). Taken together, our findings indicated that an IF regimen improved brain energy metabolism and cognitive function via restructuring gut microbiota and metabolites.

Notably, these IF-improved neuroprotective microbial metabolites are highly correlated with mitochondrial biogenesis and energy metabolism-related gene expression in the hippocampus, which could shed light on a new mechanism for how IF alleviated diabetes-induced cognitive deficits (Fig. 5). Mitochondria, as the powerhouses for eukaryotic cells, generate ATP via OXPHOS to

meet the majority of the enormous energy demands made by neurons[47]. A sufficient energy supply is required to maintain neuronal survival, neurotransmission, and the synaptic function. The fasting process also triggers AMPK activation and mTOR inactivation, and subsequently stimulates the energy metabolism and upregulates mitochondrial function to meet the energy deficiency[48,49]. We found that IF improved mitochondrial biogenesis and energy metabolism-related pathways, consistent with the upregulation of the AMPK/PGC1α pathway at the protein level (Fig. 3). Moreover, the microbial metabolites, i.e., 5-HT, TUDCA, and IPA, have been well reported to be beneficial to mitochondrial biogenesis and function. 5-HT has been reported to prevented dopamine-induced oxidative damages of mitochondria and synaptosomes[45]. As mentioned ahead, TUDCA is a potential neuroprotective agent[13,14]. It has also been reported that TUDCA treatment prevented cognitive deficits via improving mitochondrial function and reducing neuronal apoptosis in an IPA induced model of Huntington's disease model[14]. Similarly, as one of the microbial metabolites, IPA has been reported to protect against Aβ-induced neuronal death and restore mitochondrial function[15,50]. SCFAs, generated from dietary fiber by gut microbiome, has been extensively studied as potential benefits for brain health[51]. Butyrate, for instance, was found to be a potential treatment for autism spectrum disorders as its bioactivity on enhancing mitochondrial function[52]. Here, we also found treatments of microbial metabolites also improved diabetes-related cognitive deficits and increasing mitochondrial biogenesis (Fig. 6).

Furthermore, the beneficial effects of IF on cognitive function were suppressed after removing the gut microbiota in diabetic mice by antibiotics treatment (Fig. 6), suggesting a key role of microbiota composition and their derived metabolites in mediating IF- induced neuronal effects. Mitochondrial biogenesis was also downregulated by antibiotics in IF-treated mice, which further confirms that the restructured gut microbiota was highly correlated with hippocampal mitochondrial gene expression. Although there were several reports indicated that metronidazole could cross the blood–brain barrier and has potential neurotoxicity[53–55]. However, we did not observe any behavioral disorders in both db/m and db/db mice after antibiotics treatment (Fig. 6; Supplementary Fig. 7). Of note, antibiotics treatment alone also partly altered the food/water intake, insulin resistance, and cognitive behaviors in db/db mice (Fig. 6; Supplementary Fig. 8), in line with a previous report revealing that metronidazole or vancomycin treatment improves brain insulin sensitivity and behavioral changes in obese and diabetic mice[56]. Besides, antibiotics treatment reduced the generation of SCFAs and IPA, but not the levels of TUDCA and 5-HT in IF-treated db/db mice (Supplementary Fig. 9). These results indicate that the efficiency of a combined treatment of an IF regimen and antibiotics in metabolic syndrome needs to be further investigated. It is also important to further investigate the molecules and pathways that transmit microbiota changes into the brain, which could lead to the discovery of new therapeutic targets for metabolism-implicated cognitive deficits.

Overall, the current research suggests a novel and convincing link between gut microbiota and cognitive function via microbial metabolites under the treatment of an IF regimen in diabetic mice. IF treatment restructured gut microbiome and altered microbial metabolites, upregulated hippocampus mitochondrial biogenesis and energy metabolism, protected synaptic ultrastructure, and alleviated cognition and spatial memory impairments. The impact of IF on changes in microbiota/metabolites/genes should be further evaluated in clinical trials. Once validated, IF can be translated into a novel ecological approach for managing metabolic and neurodegenerative diseases.

## Methods

**Mice and antibiotics treatment**. Male BKS.Cg-Dock7$^m$ + / + Lepr$^{db}$/J (stock no: 000642) Homozygous Lepr$^{db/db}$ mice were diabetic, and heterozygous Lepr$^{db/m}$ mice were used as controls (denoted as db/db and db/m in the text) in this study. All mice were originally obtained from The Jackson Laboratory (Bar Harbor, ME) and housed in the Northwest A&F University animal facility under standard conditions with a strict 12 h:12 h light:dark cycle, humidity at 50 ± 15%, temperature 22 ± 2 °C. The animals were fed ad libitum before the IF was initiated at 4 months of age. Mice were fed a regular chow (AIN-93M, purchased from TROPHIC Animal Feed High-tech Co. Ltd, Nantong, China) and pure water.

For the first set animals, the db/m mice and the db/db mice were divided into two subgroups ($n = 10$), including AL feeding (db/db) and IF (db/db-IF), wherein the IF mice were deprived of food for 24 h, every other day and were fed ad libitum on the intervening day for 28 days (Fig. 1a). Bodyweight, food intake, and water consumption were recorded on fasting day. These animals performed behavioral tests, and then were killed to collect serum and tissues.

The second set of mice (same grouping with first set, $n = 10$–11) were treated with the same IF regimen schedule and then were killed to collect the serum, hippocampus, gut, and fecal samples for multi-OMICS study. Fecal, serum, and hippocampus samples were collected to analyze gut microbiome, metabolome, and RNA sequencing, respectively.

The third set of animals were divided into eight groups, including db/m ($n = 10$), db/m-IF ($n = 10$), db/m-antibiotics ($n = 10$), db/m-antibiotics-IF ($n = 10$), db/db ($n = 7$), db/db-IF ($n = 13$), db/db-antibiotics ($n = 7$), db/db-antibiotics-IF ($n = 13$). The different sample size was due to requirement to distinguish the impact of IF on feeding and fasting day. The IF regimen schedule was the same as the previously mentioned sets. The antibiotics cocktail (Penicillin G sodium 0.4 g L$^{-1}$, metronidazole 0.4 g L$^{-1}$, neomycin sulfate 0.4 g L$^{-1}$, streptomycin sulfate 0.4 g L$^{-1}$, vancomycin hydrochloride 0.25 g L$^{-1}$ for db/db mice, half concentration of same antibiotics for db/m mice) were given in the drinking water starting 14 days before the IF regimen and throughout the experiment (Fig. 6a). The 16S rRNA copies in the feces of the animals after antibiotics treatment were detected with qPCR (Fig. 6b), as previous described[57]. All the behavior studies and biochemical samples were collected/detected on the fasting day of IF regimen. One mouse in the db/db group was drowned during the water-maze test, and relevant data after that were excluded. For plasma metabolomics, two samples were excluded due to the poor quality of measurements.

The fourth set of animals were divided into six groups ($n = 8$), including db/m, db/db, db/db-IPA, db/db-5-HT, db/db-TUDCA, db/db-SCFAs. For db/db-IPA, db/db-5-HT, db/db-TUDCA groups, mice were intraperitoneally injected with IPA (10 mg kg$^{-1}$ d$^{-1}$), 5-HT (1 mg kg$^{-1}$ d$^{-1}$), and TUDCA (250 mg kg$^{-1}$ d$^{-1}$) dissolved in saline, respectively. The mice in other groups were injected with same volume saline. For the SCFAs group, the SCFAs (acetate 67.5 mM, propionate 40 mM, butyric acid 25 mM) were dissolved in the drinking water of the mice. The mice were treated for 14 days. The treatment was also performed during the behavioral tests.

After IF regimen, the antibiotics treatment, and metabolites treatment, the cognitive behavioral assessment was determined with a water-maze test. After behavioral tests, mice were killed, then the serum and tissues samples were collected by either snap-frozen by liquid nitrogen and store at −80 °C or directly stored in 4% paraformaldehyde for histological analysis.

All of the experimental procedures were followed using the Guide for the Care and Use of Laboratory Animals: Eighth Edition (ISBN-10: 0-309-15396-4). We have complied with all relevant ethical regulations for animal testing, and research and protocols were approved by the Northwest A&F University, and BGI Institutional Review Board on Bioethics and Biosafety (BGI-IRB).

**Insulin-tolerance tests and analyses of plasma contents**. Insulin-tolerance tests protocol was modified from our previous research[58]. The mice were fasted for 6 h before the tests. The insulin (0.75 U kg$^{-1}$, Sigma Aldrich, USA) was injected, and blood glucose was measured before (0 min) and after the injection (15, 30, 60, 120 min) using a OneTouch® SelectSimple™ glucometer (LifeScan Inc., China). For other content analysis in plasma, all reagents and kits source and identifiers were listed in Supplementary Table 1. The plasma insulin, leptin, 5-HT, and LPS were detected using ELISA kits purchased from Xinle Bio Co.,Ltd., Shanghai, China. The homeostasis model assessment of insulin resistance (HOMA-IR) was calculated as (fasting insulin concentration (mU L$^{-1}$) × fasting glucose concentration (mg dL$^{-1}$) × 0.05551)/22.5.

**H&E and immunochemistry staining**. For H&E staining, the adipose and gut (colon) tissues were embedded in paraffin for staining with hematoxylin and eosin. The adipocytes sizes were measured by Image J (developed by Wayne Rasband from NIH, USA) software with Adipocytes Tools plugins after pictures were recorded by light phase contrast microscopy. The immunohistochemical staining was performed according to previous study[59]. The fixed brain and gut sections were exposed to the primary antibodies at 4 °C overnight; the information of primary antibodies is shown in the Supplementary Table 1. After incubation, the sections were washed three time with PBS and incubated with biotinylated goat anti-rabbit or goat anti-mouse diluted in a secondary antibody dilution buffer.

After staining nuclei by hematoxylin, neutral resin was used for sealing the sections. Then, the IHC staining images were obtained using an inverted fluorescent microscope (×400).

**Morris water-maze tests**. The Morris water maze is one of the most widely used tests in behavioral neuroscience for studying the psychological processes and neural mechanisms of spatial learning and memory. The protocol was modified from our previous research and detailed described as following[60]. The apparatus consists of a large circular pool, 1.5 meters in diameter and a height of 35 cm (XR-XM101, Shanghai Xinruan Information Technology Co. Ltd, Shanghai, China), containing water at around 25 °C. The mice received four habituation trainings on day 0. The platform was visible (2 cm above the water surface), and the water was un-dyed. The water was then made opaque by adding white-dye (food-grade titanium dioxide) that helped to hide the submerged platform (day 1–day 6). Test trials were conducted for 5 consecutive days (day 1–day 5). On day 6, a probe trial was conducted in which mice were placed in the pool for 60 s without the platform and the time that was spent in the target quadrant, the latency to the platform and the number of platform crossings were measured. All the data were recorded automatically using a video tracking system (SuperMaze software, Shanghai Xinruan Information Technology Co., Ltd, China).

**Electron microscopy for the structural analysis of the hippocampus**. A transmission electron microscope (TEM) analysis was done after the collection of CA1 region of hippocampus. The hippocampus was split and treated in a cold fixative solution made of 2.5% glutaraldehyde (pH 7.2) at 4 °C for 4 h. After washing with PBS (0.1 M, pH 7.2) thrice. Then the specimens were post-fixed in 1% OsO$_4$ (in 0.2 M PBS, pH 7.2) at 4 °C for 1 h and washed again with PBS (0.1 M, pH 7.2) thrice. The specimens were dehydrated for 15–20 min each in a graded series of ethanol solutions (30, 50, 70, 80, 90, and 100%) and then transferred to acetone for 20 min incubation. Materials were then permeated in an acetone–resin mixture (1:1) for 1 h at 25 °C and then transferred to an acetone–resin mixture (1:3) overnight. Ultrathin sections were placed in the regions which were closed to the embedded blocks and kept away from the dorsal rim area, stained with uranyl acetate and alkaline lead citrate for 15 min, and then observed using JEM-1230 TEM (JEOL, Tokyo, Japan) at 80 kV and acquired using a side-inserted BioScan Camera (Veleta, EMSIS GmbH, Germany).

**Ussing chamber assay**. Ussing chambers are a common tool to evaluate the gut barrier ex vivo. Directly after dissection of the intestine, 1.5 cm pieces of the jejunum (the center of the small intestine) were opened along the mesenteric border and mounted as flat sheets in the Ussing chambers separating the chamber into two halves (BeiJing KingTech technology Co.Ltd, Beijing, China). Luminal and serosal surfaces were continuously exposed to carbogen-gassed Krebs buffers (CaCl$_2$·2H$_2$O 7.35 g, NaCl 13.67 g, KCl 7.01 g, NaHCO$_3$ 4.2 g, MgCl$_2$·6H$_2$O 4.88 g, glucose 3.96 g, NaH$_2$PO$_4$·2H$_2$O 3.74 g, dissolved in 2 L ddH$_2$O, pH 7.4) at 37 °C. Tissues were equilibrated for 45 min in the presence of 0.09 g L$^{-1}$ fluorescein on the luminal side. For permeability measurements, the fluorescence intensity of the serosal buffer was determined at 0, 15, 30, 45, and 60 min and used to calculate permeability as expressed in gradient.

**Western blots**. The protein of gut and brain tissue were extracted using a protein-extraction reagent. The total tissue proteins ($n = 3$) were separated by SDS-polyacrylamide gel electrophoresis (SDS-PAGE), and then transferred onto a polyvinylidene fluoride (PVDF) membrane by using a wet transfer apparatus. Appropriate antibodies were used and the immunoreactive bands were visualized with an enhanced chemiluminescence reagent. The information of primary antibodies is shown in Supplementary Table 1. Quantification of the western blots results using the band densitometry analysis was performed with Quantity One software.

**qRT-PCR**. The total RNA extracted from frozen tissues using TRIzol reagent (Jingcai Bio., Xi'an, Shaanxi, China) was determined and reverse transcribed for real-time PCR. Relative mRNA expression was quantified using SYBR Green dye (TB Green Premix Ex Taq II) and specific primers. Real-time PCR was carried out in a CFX96TM real-time system (Bio-Rad). The following conditions were used: 95 °C for 10 min, then 95 °C for 15 s, and 60 °C for 1 min in 40 cycles. The 2$^{-\Delta\Delta CT}$ method was used to analyze the relative changes in gene expression. For the mitochondrial biogenesis, the total DNA extracted from brain tissue using a DNA extraction kit (Bioteke Co., Beijing, China) were also determined by real-time PCR. The mitochondria number were indicated by the lower ratio of mitochondrial DNA (mtDNA, COX2) to nuclear DNA (nDNA, globin). Reagent information and primer sequences are shown in Supplementary Table 1.

We filtered and trimmed the reads using Trimmomatic v0.38. Clean reads were mapped to the *Mus musculus* genome sequence (ftp://ftp.ncbi.nlm.nih.gov/genomes/all/GCF/000/001/635/GCF_000001635.26_GRCm38.p6) using Hisat2 v2-2.1.0. The reads of each sample were then assembled into transcripts and compared with reference gene models using StringTie v1.3.4d. We merged the 31 transcripts to obtain a consensus transcript using a StringTie-Merge program. Transcripts that did not exist in the CDS database of the Mus musculus genome were extracted to

predicted new genes. The gene expression FPKM values were calculated using StringTie based on the consensus transcript. DEG analysis was performed using Ballgown v2.12.0, an R programming-based tool designed to facilitate flexible differential expression analysis of RNA-Seq data. Only genes with fpkm >1 ($n = 10$) were subjected to analysis and the differential expression genes was determined (FDR-$p < 0.05$).

An unsupervised co-expression network analysis of all genes was performed using R package WGCNA v1.64. The signed scale-free topology overlap matrix was computed, and co-expression modules were defined from this network. For each identified module, the hub genes were defined by module connectivity (Pearson's correlation >0.8) and correlations between each intra-module gene and treatments (correlation >0.85). The co-expression network was visualized using the Cytoscape. The GO and KEGG pathways were annotated using WebGestalt (http://www.webgestalt.org/2019/) (false discovery rate FDR-adjusted $p < 0.05$). A detailed description of data processing and analysis is provided in Supplementary Information.

**16S rRNA Microbiome sequencing**. Fecal samples were collected from respective groups either at the beginning or before the behavioral tests. The total cellular DNA was extracted with the E.Z.N.A. Stool DNA Kit (Omega) according to the company instructions. The bacterial hypervariable V3–V4 region of 16S rRNA was chosen for MiSeq (Illumina, CA, USA) paired-end 300 bp amplicon analysis using primer: 341_F: 5′-CCTACGGGNGGCWGCAG-3′ and 802_R: 5′-TACNVGGGTATCTAATCC-3′. The library preparation followed the method published previously[61].

The raw reads were merged and trimmed, following by removal of chimera and constructing zero-radius Operational Taxonomic Units (zOTUs) with UNOISE implemented in Vsearch (v2.6.0). The green genes (13.8) 16S rRNA gene database was used as a reference for annotation. Detailed algorithms and parameters are given in Supplementary Information. All the samples were rarified to 28,257 counts for alpha diversity index calculation. The raw OTU table was normalized with cumulative sum scaling[62] (CSS) to calculate Bray Curtis, unweighted and weighted Unifrac distance, followed by a permutation test (Vegan:adonis) to detect differences among intervention groups. Constrained analysis of principal coordinate (CAP, R package "vegan") was applied to identify the influence of mice gene type and IF on microbiota and time was introduced as a partial term to remove the background effect of time. CSS-normalized OTU data were used to calculate relative abundance and summarized in different levels. Specific taxa comparisons among groups were analyzed using the analysis of composition of microbiomes (ANCOM). FDR-$p < 0.05$ was considered to be a significant difference. We also computed Pearson correlations between centered log-ratio transformed relative abundance of genera and bodyweight, blood glucose, food intake, water intake, LPS, leptin, GABA, 5-HT, insulin, and fecal SCFAs. Rarified OTU data were used to predict functional gene with PICRUSt (v1.1.3). Predicted gene was annotated with KEGG at different levels, and the significantly abundant pathways were identified by edgeR with FDR-$p < 0.1$.

**SCFAs analysis**. The concentrations of SCFAs (acetate, propionate, and butyrate) were determined with a Shimadzu GC-2014C gas chromatograph (Shimadzu Corporation, Kyoto, Japan) equipped with a DB-FFAP capillary column (30 m × 0.25 μm × 0.25 mm) (Agilent Technologies, Wilmington, DE, USA) and flame ionization detector. Approximately 200 mg of the fecal content sample was homogenized with 1 ml of distilled water; then, 0.15 mL of 50% H2SO4 (w/w) and 1.6 mL of diethyl ether were added. After the samples were incubated at 4 °C for 30 min, they were centrifuged at 8000 rpm for 5 min. The organic phase was collected and analyzed using gas chromatography as follows. The initial temperature was 50 °C, which was maintained for 3 min and then raised to 130 °C at 10 °C per min, increased to 170 °C at 5 °C per min, increased to 220 °C at 15 °C per min and held at this temperature for 3 min. The injector and the detector temperature were 250 °C and 270 °C, respectively.

**Untargeted metabolomics**. Plasma samples were collected after the animals were killed, and were stored at −80 °C until analysis. Samples were analyzed using an ultra-performance liquid chromatography (UPLC) system and a high-resolution tandem mass spectrometer Xevo G2 XS QTOF (Waters, UK). Reverse-phase chromatography was employed, using both positive and negative electrospray ionization modes (ESI, RP+ and RP−). A 10 μl of the sample solution was injected on an ACQUITY UPLC HSS T3 column (100 mm × 2.1 mm, 1.8 μm, Waters, UK). The column oven was maintained at 50 °C. The flow rate was 0.4 ml per min and the mobile phase consisted of solvent A (water + 0.1% formic acid) and solvent B (acetonitrile + 0.1% formic acid). Gradient elution conditions were set as follows: 0–2 min, 100% phase A; 2–11 min, 0% to 100% B; 11–13 min, 100% B; 13–15 min, 0% to 100% A. ESI source was operated using the following conditions: for the positive ion mode, the capillary and sampling cone voltages were set at 3.0 kV and 40.0 V, respectively. For the negative ion mode, the capillary and sampling cone voltages were set at 2.0 kV and 40.0 V, respectively. The mass spectrometry data were acquired in Centroid MSE mode. The TOF mass range was from 50 to 1200 Da, and the scan time was 0.2 s. For the MS/MS detection, all precursors were fragmented using 20–40 eV, and the scan time was 0.2 s. During acquisition, the LE signal was taken every 3 s to calibrate the mass accuracy.

Samples were analyzed in one batch with a randomized injection order. The stability and functionality of the system was monitored throughout all the instrumental analyses using quality controls, i.e., the pooling of all samples acquired at the beginning of analytical sequence and after every ten injections. Data preprocessing was performed using Progenesis QI (version 2.2). In total, 6295 and 6893 metabolic features were detected. A support vector regression-based normalization was performed to minimize unwanted variations in feature intensities, resulting 5604 and 5230 features in RP+ and RP−, respectively, with relative standard deviations below 20%[63]. They were considered as qualified features and were subjected to statistics. P-values for fold change were adjusted for multiple testing using Benjamini–Hochberg false discovery rate (FDR). Principal component analysis was performed on auto-scaled intensities (mean = 0, standard deviation = 1) of all quantified metabolite features detected in RP+ and RP−, respectively using R package "mixOmics".

Metabolite identification was carried out based on accurate mass and product ion spectrum matching against online databases and literature. The list of microbial cometablites (i.e., metabolites whose levels were modified by gut microbiota) was determined according to Rowan et al.[12]. Annotated microbial metabolites are provided in Source data file of Fig. 4.

**Integrated multi-OMICS analysis**. Multivariate predictive modeling on each omics data set was conducted using partial least square-discriminant analysis incorporated into a repeated double cross-validations framework (rdCV-PLSDA)[34]. Outperforming the standard cross-validation, the double cross-validations procedure separates cross-validations into an outer "testing" loop, and an inner "tuning" (or validation) loop to further reduce bias from overfitting models to experimental data. To gain a robust and reliable estimate of model performance, 200 repetitions of the outer cross-validations loop was performed, followed by permutation analysis (n = 1000)[34,64].

A multivariate dimension reduction method, DIABLO (Data Integration Analysis for Biomarker discovery using a Latent component method for Omics), was employed for multiple omics integration[33]. A random use of full design matrix was applied to look for linear combinations of variables from each omics data set that are maximally correlated. A tuning procedure was applied to determine the optimal number of key variables in each data set to be selected with a minimum misclassification rate. Model performance was then evaluated by tenfold cross-validation. The detailed step-by-step workflow of DIABLO analysis is provided in Supplementary Information.

**Statistical analysis**. Other than RNA sequencing, gut microbiome, and metabolome data, other data were reported as mean ± SEM, significant differences between mean values were determined by Student's t test and one-way ANOVA. The data for antibiotics treatment experiments were determined with two-way ANOVA with IF and antibiotics as factors. A post hoc test was performed using Tukey's test for multiple comparison test by Graphpad Prism 6.0 software. Other software information employed in this study is deposited in Supplementary Information. The measurements were taken from distinct samples. Means were considered to be statistically significant, if p < 0.05.

**Reporting summary**. Further information on research design is available in the Nature Research Reporting Summary linked to this article.

## Data availability

The source data file underlying Figs. 1–6 and Supplementary Figs. 1–10 ware provided as a Source data file and some OMICs data used in these figures were also deposited as Supplementary Data 1–10. The accession number for the entire RNA-seq data set reported in this manuscript is GSE125387. The accession number for the entire 16 S rRNA sequencing data set reported in this manuscript is SRP181000. The RNA-seq and 16S rRNA sequencing data that support the findings of this study have been also deposited in the CNSA (https://db.cngb.org/cnsa/) of CNGBdb with accession number CNP0000608. Raw data were also deposited in figshare data set as follows: https://figshare.com/s/58ccac4aa614dd4ade84.

## Code availability

All the software and R programming codes used in current study are ether commercial or have been previously published. Detailed information is provided in the text and in Supplementary methods. No custom code has been developed and applied in current study. A step-by-step R markdown file for integrative modeling is also provided in Supplementary Information, together with the multi-OMICs data set.

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

## Acknowledgements

This work was financially supported by the National Key Research and Development Program of China (No. 2016YFD0400601), National Natural Science Foundation of China (81803231), National Natural Science Foundation of China (81871118), General Financial Grant from China postdoctoral science foundation (2016M602867), Special Financial Grant from China postdoctoral science foundation (2018T111104), and Fundamental Research Funds for the Central Universities (2452017141) supported this research. We also thank Prof. Rikard Landberg (Chalmers University of Technology, Sweden) for great comments and suggestions to improve the work.

## Author contributions

Z.L., H.Z., R.S., W.Z., Y.H., X.J., L.W., Q.W., D.W., B.R., J.W. (Jia Wang), X.T., T.Z., X.L. (Xiaoning Liu), T.Y., C.C., L.L. and L.S. performed the experiments and analyzed the data, Z.L., L.S., X.D., S.Z. and X.L. (Xuebo Liu) designed the study, Z.L., Y.H., X.J., F.Y., E.C., L.S. and X.D. wrote the paper, Z.L., H.Z., R.S., J.W. (Jiamin Wang), J.P. and L.S. prepared the figures. All authors discussed the results and commented on the paper.

## Competing interests

The authors declare no competing interests.
