## [Peer Review File · Nature Communications]

Reviewers' comments:

Reviewer #1 (Remarks to the Author):

Liu and colleagues assess the effects of intermittent fasting (IF) on diabetes through a multi-omics approach. The authors conclude that IF enhances hippocampal mitochondrial biogenesis and metabolism-related gene expression, alters the composition of the gut microbiota and improves metabolic phenotypes that correlate with cognitive behavior. Additionally, the authors find that depletion of the gut microbiota through antibiotic treatment abolishes the effect of IF on hippocampal metabolism and cognitive behavior. Finally, the authors conclude a 28-day IF treatment alleviates diabetes-induced cognitive impairment via a microbiota-metabolites-brain axis. This work draws on previous studies demonstrating that IF-induced changes in the gut microbiota mediates effects on host metabolism (Li et al *Cell Metabolism* 2017), that the gut microbiota changes hippocampal mRNA expression (Chen et al *Behav Brain Res* 2017, Clarke et al *Molecular Psychiatry* 2017, Diaz-Heijitz et al *PNAS* 2011, Frohlich et al *Brain Behav Immun* 2016), and that antibiotic treatment affects hippocampal function (Frohlich et al *Brain Behav Immun* 2016). The novelty of this paper lies in furthering these studies to study effects of IF-induced changes in the gut microbiota on cognitive behavior and hippocampal metabolic function. Conceptually, the paper will be of interest to others in the field because information for how the gut microbiota impacts cognition is currently lacking and multi-omics studies in this area of study are limited. However, the manuscript's clarity can be improved as suggested below, and while most claims are supported by the paper, the major claim that microbial metabolites impact cognitive behavior and metabolism lacks sufficient experimental evidence. Overall, the manuscript would be more promising if the authors identified particular microbiota-associated metabolites that demonstrably mediate the behavioral and neurophysiological changes they describe in the antibiotic-treated and fasted mice.

Major comments/concerns:

1) For figure 3A, if the rationale for gene selection is that IF is correcting the db/db metabolic impairment phenotype to the level of the db/m controls, what is the logic behind selecting the genes that specifically increase for only the fasting group and relative to both db/db and db/m mice? Would it be more informative to select those that are similar in expression in both the fasted and the db/m groups when compared to the db/db mice? A comparison of this nature could identify genes that are restored to control levels during fasting.

2) When Fig. 4H is addressed on page 19, it is not clear that the data represent inferred metagenomic data rather than authentic metagenomic data. The authors should clarify this and discuss the caveats and limitations of the inferences. While PICRUSt may yield pathways of interest to probe, this should either be validated by another assay concretely measuring gene expression in these pathways or by measuring related metabolites like bile acids.

3) For Figure 6D, the authors did not address whether the antibiotic-induced changes in weight gain were due to increased cecal size. It will be important to determine how much of the weight gain in this group is truly a difference in fat mass rather than overall weight. Also how did antibiotic treatment affect other parameters for metabolism in this case, like HOMA-IR, fasting insulin, and glucose levels (as measured in Fig. 1)?

4) The authors demonstrate that there are metabolites that associate with the improved cognitive behavior seen in fasted db/db mice and that the microbiota is required for the effect (seen with antibiotic depletion model). However, it will be important to directly test the role of these metabolites in the animal model to evaluate causation vs. correlation. The author's claim in the discussion of demonstrating "for the first time that a 28-day IF treatment alleviated diabetes-induced cognitive impairment via a microbiota-metabolites-brain axis" is not completely proven with such evidence. For example, since the authors previously identify short chain fatty acids as microbiota-associated metabolites of interest (Fig. S4G-R), the authors could test whether short chain fatty acids are reduced in the fasted antibiotic-treated group and if so, administer short chain fatty acids and test cognitive behavior in this group.

5) Many methods are stated as being "performed as previously described". The authors should consider adding additional methodological details in the methods section to enhance clarity.

Minor comments/concerns:

1) The authors should include more specific contextual information regarding prior literature in this area. For example, on pg. 4, the authors state, "They are also strongly linked to whole body metabolism and function particularly, and have been shown to have potential neuroprotective effects". Which metabolic effects has the gut microbiota been connected to previously and which types of neuroprotective effects?

2) Please clarify on page 18 that claudin-1 is a tight-junction protein, which is how this directly relates to intestinal permeability.

3) On page 18: "Gut microbiota was determined" specifically means "gut microbiota composition was determined..." and that this was performed by 16S rDNA sequencing. Also, it needs to be specified that while alpha diversity did not change here (Fig. 4B), beta diversity did (Figure 4C) (here the text simply states "microbiota diversity", but this should be clarified).

4) There are many grammatical errors throughout the manuscript that require revision: page 2 "...circulated metabolites that were affect by IF", page 18, "goblet cells numbers" should be changed to "goblet cell numbers". Pg. 4: "the gut microbiota plays a vital role in inter-phasing" page 25, "A DIABLO integrative modeling was then built on the abovementioned..."

Page 31: "complications via re-structuring"

5) Because the scale in Fig. 1N-P varies from graph to graph it is difficult to cross-compare groups. I think clarity would be enhanced here by either displaying the data separately with the same scale on

the y-axis or by plotting all groups on one graph. Otherwise, it is difficult to assess the claim that the db/db-IF group “significantly decreased...the size of adipocytes” as stated on pg. 7.

6) In Figure 1, the figure legend is inaccurate, where for example part I is listed as “fasting insulin level”, but the graph is for HOMA-IR, part J is listed as “fasting glucose”, but is actually leptin, part K is listed as HOMA-IR, but is actually 5-HT, etc.

7) On page 10 when the following sentence is stated, “We also observed that the expression of BDNF, a neurotrophic factor that plays an essential role in maintaining neural survival and synaptic function, was increased”, this should be specified to state what the increase is relative to. A similar improvement can be made to the two sentences directly following this.

8) The title for Figure 2 can be improved to include a description of the signaling pathways that were altered with intermittent fasting.

9) In Figure 2, part I is incorrectly labelled in the legend as “G”.

Reviewer #2 (Remarks to the Author):

The study by Liu et al. describes the effects of intermittent fasting (IF) on the gut microbiome and cognitive capacity in a murine model of type 2 diabetes. There currently is a growing interest in understanding the apparent beneficial effects of fasting in relation to different disease phenotypes from a mechanistic point of view. The study of Liu et al. provides important results which further our understanding of the interrelationship between fasting, microbiome-driven metabolism and molecular effects in the brain. The study is therefore both timely and important. In general and according to my assessment, the work has been performed to a high standard. Nevertheless, I have a few reservations which may preclude publication of the manuscript in its current form.

Major comments

1. Although the authors have demonstrated significant shifts in the gut microbiome, circulating metabolites and gene expression in the brain following IF and have linked these changes to improvements in cognitive function, the study fails to conclusively prove causal relationships. Granted, the administration of antibiotics and the resulting reduction in microbiome diversity and activity does provide some inroads into untangling causality but, given that the authors have identified specific microbiome-derived molecules (e.g. bile acids, serotonin, IPA, etc.) which likely confer the observed effects, I am missing the final set of experiments with these molecules to prove at least one mechanistic connection. Therefore, I would suggest that the authors add one

experiment by which they would prove that fasting-induced changes to specific metabolites result in the expression changes in the brain and linked cognitive phenotypes.

2. The figures are extremely dense (see especially Figures 1, 3 & 4) and largely illegible. I suggest the authors revise their figures and determine what information is really essential and/or which can be moved to the supplement.

3. The authors discuss statistically significant differences in the text without highlighting which tests were used and without including corresponding P values. The authors should rectify this throughout the manuscript.

4. The text is at times difficult to follow and comprehend. It might be helpful to have it revised by a native English speaker. I make a few corresponding suggestions below.

Minor comments

P4: "interphasing"; "linking" might be the more appropriate term.

P4: "inter-individual variation".

P5: "reported to be beneficial in the context of metabolic syndrome"

P5: "Intermittent fasting (IF) regimens represent a periodic"

P5: "via modulation of"

P5: "under-explored"

P5: "gut microbiota plays a pivotal"

P7: "compared to ad libitum fed"

P9: "Data presented as"

P14: "hippocampi was"

P14: "most of which were enriched"

P14: "via up-regulating mitochondrial genes"

P14: Is "brown" the correct term to use?

P15: "hubgenes were enriched"

P15: "protein expression"

P17: "C Network and GO annotation"

P18: "16S rRNA gene v3-v4 amplicon"

P18: "murine genotype were still"

P19: "Allobaculum and Bifidobacterium"

P20: "was placed on a a priori"

P20: "ad libitum feeding"

P25: "(Fig. 5C, Fig. S5B) and"

P27: "cross-validations"

P27: "plotDiablo function"

P28: "Microbiota are"

P28: "affected by the reduction of the microbiota"

P28: Include details on the antibiotic treatment

P31: "via restructuring of the gut microbiota"

P32: "An integrated analysis"

P33: "enrichment in Lactobacillus"

P34: "restructuring"

P35: "neuronal survival"

P51: "Statistical analysis"

Dear Reviewers,

We sincerely appreciate for your constructive and helpful comments and suggestions to our manuscript entitled "**Gut Microbiota Mediates Intermittent-Fasting Alleviation of Diabetes-Induced Cognitive Impairment**".

We have tried our best to address the comments and questions raised by the reviewers and the Editor. Overall, we have now conducted additional animal studies to evaluate the specific role of gut microbiota related metabolites in the observed phenotypes, i.e. cognitive function and mitochondrial biogenesis in db/db mice. We have also extensively revised our manuscript in order to clarify and highlight the original findings, as well as the conclusions we made. In addition to the transcriptomics and 16s RNA sequencing data that were uploaded in the registry (GSE125387 and SRP181000, respectively), we have now provided a dataset including the metabolites, OTUs and genes that were of great interest relevant for the beneficial impact of intermittent-fasting on phenotypes and a detailed description of multi-omics integrative analysis. This information could the scientific community to easily reproduce results of our work.

In addition to reviewers' comments, to improve the accuracy and predictive performance of multivariate modeling for multi-omics data, we have made changes in the modelling parameters:

1. We performed 200 repetitions for repeated double cross validation-

PLS modelling, instead of using 100.

2. We apologize for the lack of propionate data in previous analysis. We have now carefully checked the data and reperformed repeated double cross validation- PLS modelling and integrative analysis. Data and R script used for integrative analysis are provided in Supplementary materials

All changes are highlighted in red in the revised version of the manuscript.

The major corrections/revisions are listed below, point-by-point:

Reviewer #1 comments:

Liu and colleagues assess the effects of intermittent fasting (IF) on diabetes through a multi-omics approach. The authors conclude that IF enhances hippocampal mitochondrial biogenesis and metabolism-related gene expression, alters the composition of the gut microbiota and improves metabolic phenotypes that correlate with cognitive behavior. Additionally, the authors find that depletion of the gut microbiota through antibiotic treatment abolishes the effect of IF on hippocampal metabolism and cognitive behavior. Finally, the authors conclude a 28-day IF treatment alleviates diabetes-induced cognitive impairment via a microbiota-metabolites-brain axis. This work draws on previous studies demonstrating that IF-induced changes in the gut microbiota mediates effects on host metabolism (Li et al Cell Metabolism 2017), that the gut microbiota changes hippocampal mRNA expression (Chen et al Behav Brain Res 2017, Clarke et al Molecular Psychiatry 2017, Diaz-Heijitz et al PNAS 2011,

Frohlich et al Brain Behav Immun 2016), and that antibiotic treatment affects hippocampal function (Frohlich et al Brain Behav Immun 2016). The novelty of this paper lies in furthering these studies to study effects of IF-induced changes in the gut microbiota on cognitive behavior and hippocampal metabolic function. Conceptually, the paper will be of interest to others in the field because information for how the gut microbiota impacts cognition is currently lacking and multi-omics studies in this area of study are limited. However, the manuscript's clarity can be improved as suggested below, and while most claims are supported by the paper, the major claim that microbial metabolites impact cognitive behavior and metabolism lacks sufficient experimental evidence. Overall, the manuscript would be more promising if the authors identified particular microbiota-associated metabolites that demonstrably mediate the behavioral and neurophysiological changes they describe in the antibiotic-treated and fasted mice.

Response:

We thank the comments and suggestions made by the reviewer and we certainly agree. We have now conducted additional experiments to assess the role of certain microbiota-associated metabolites in improving cognitive behavior and metabolism. We have also extensively revised our manuscript to improve the clarity and highlight the novelty of our work. Detailed responses are shown as follows:

Major comments/concerns:

Comment 1: For figure 3A , if the rationale for gene selection is that IF is correcting the db/db metabolic impairment phenotype to the level of the db/m controls, what is the logic behind selecting the genes that specifically increase for only the fasting group and relative to both db/db and db/m mice? Would it be more informative to select those that are similar in expression in both the fasted and the db/m groups when compared to the db/db mice? A comparison of this nature could identify genes that are restored to control levels during fasting.

Response:

We thank for questions raised by the reviewer. We try to clarify:

We mainly focused on the genes that were specifically increased after IF treatment compared to db/db and db/m mice, based on both hypothesis-driven strategy and data-driven strategy. Specifically, we hypothesized that substantial differences could be observed in the expressions of energy metabolism-related genes in db/db mice brain compared to those of the db/m mice, since the db/db mice has a severe insulin resistance in central nervous system (**Fig.2**). As expected, we found that most of genes specifically increased in IF group are highly related to mitochondrial function and energy metabolism, which may partly explain the impact of IF on reversing the insulin resistance in db/db brain (which was normal in db/m mice).

Moreover, by using the data-driven strategy, results from the weighted correlation network analysis (WGCNA) clearly showed that the WGCNA-derived “brown” module was positively correlated with the IF regimen ($r = 0.892$, $p = 2e-11$) and it consisted of 1,044 genes (88.40%) identified using DEG (group1, genes highly expressed in IF group). This indicates that genes upregulated by IF, mainly involved in energy metabolism, play the core role of IF in improving db/db mice brain function.

We definitely agree with reviewer that genes that may be restored to control levels during fasting are of great importance. We found that expressions of 483 genes in db/db mice hippocampus were comparable to db/m mice level after IF treatment (304 of which were higher in db/db mice, and 179 of which were lower in db/db mice). These genes are listed in group 3 and group 6 in **Supplemental spreadsheets S3**. The KEGG analysis of these genes were also performed. The genes marked in blue were down-regulated in db/db mice but were improved by IF treatment (**Figure S3B**), which are also functionally involved in mitochondrial function and energy metabolism. Thus, these results did not influence the conclusion we made.

We have now revised results section accordingly. However, in order to avoid complexity of this manuscript, we still prefer to mainly highlight the identified genes increased after IF treatment compared to db/db and db/m mice.

The related result section has been revised as following, **Page 12**:

Among them, 1,181 genes were found to be highly expressed in db/db-IF mice compared to db/db and db/m mice, using Differentially Expressed Gene (DEG) analysis (DEG-group 1 genes) (FDR-p < 0.05) (Fig.3A, Supplemental spreadsheets S3), most of which were enriched in mitochondrial-related GO terms. IF also elevated genes related to the KEGG pathway of oxidative phosphorylation (OXPHOS) via up-regulating mitochondrial genes expression (Fig. 3B). Moreover, the expressions of 483 genes in db/db mice hippocampus were comparable to those levels in db/m mice after IF treatment (FDR-p < 0.05) (Supplemental spreadsheets S3). The KEGG analysis of these genes were also indicated that IF treatment improved energy metabolism-related genes expressions in OXPHOS that were down-regulated in db/db mice (Fig. S3B).

Comment 2: When Fig. 4H is addressed on page 19, it is not clear that the data represent inferred metagenomic data rather than authentic metagenomic data. The authors should clarify this and discuss the caveats and limitations of the inferences. While PICRUSt may yield pathways of interest to probe, this should either be validated by another assay concretely measuring gene expression in these pathways or by measuring related metabolites like bile acids.

Response:

Thanks for reviewer's comments and we agree. In this work, we conducted the PICRUSt analysis in order to get some hints for the functional

capability of the gut microbiota. By means of ancestral-state reconstruction algorithm, PICRUSt utilizes the marker gene data and a database of reference genomes to estimate composite metagenome (Langille, Morgan GI, et al. Nat. Biotechnol. 2013).

We totally agree with reviewer that PICRUSt has its autologous defects on account of the availability of appropriate reference genomes and undetectable resolution of 16s rRNA sequences on strain variation. Like authentic metagenome, PICRUSt can only provide the gene or pathway enrichment rather than transcription or translation. We applied this analysis aiming to provide a possible functional profiling of gut microbiome, which has not been integrated into multi-omics analysis. Moreover, our conclusions regarding the role of IF in influencing levels of bile acids in the host were based on metabolome analysis (**Fig. 4G**).

We have now revised the related description in the result section, to improve the clarity and the limitations using the PICRUSt have now been provided in the Discussion section as following:

Result section Page 17:

*Meanwhile, a PICRUSt analysis revealed 11 differently abundant KEGG gene pathways (FDR-p < 0.1). Among them, the primary and secondary bile acid biosynthesis were abundant in the db/db-IF group (**Fig. 4F, Supplemental spreadsheets S8**).*

Discussion section Page 34:

However, PICRUSt has its autologous defects on account of the availability of appropriate reference genomes and undetectable resolution of 16s rRNA sequences on strain variation. In the current study, using metabolome analysis, we found that IF increased plasma levels of BAs including CA, TUDCA, MCA, and DCA and these BAs were correlated with IF affected gut microbiome and transcriptome in the hippocampus.

Comment 3: For Figure 6D, the authors did not address whether the antibiotic-induced changes in weight gain were due to increased cecal size. It will be important to determine how much of the weight gain in this group is truly a difference in fat mass rather than overall weight. Also how did antibiotic treatment affect other parameters for metabolism in this case, like HOMA-IR, fasting insulin, and glucose levels (as measured in Fig. 1)?

Response:

We appreciate the questions proposed by reviewer and we have revised the results section to highlight our findings. We analyzed the cecal size and other organs weight of mice in different groups. And we found that the antibiotic-induced bodyweight changes might partly due to the increased cecal size. Besides, the liver weight of antibiotics-IF group was also significantly higher than the IF treatment group. But there was no difference on the eWAT weight between the antibiotics treatment and IF treatment group. We also added the results of the impacts of antibiotic treatment on HOMA-IR, fasting

insulin, or fasting glucose levels. It has been found that antibiotic has no impact on the beneficial effects of IF on insulin resistance. The antibiotic treatment alone also significantly reduced the HOMA-IR value of db/db mice. As we indicated in the Discussion section, the efficiency of a combined treatment of an IF regimen and antibiotics in metabolic syndrome needs to be further investigated. These results are presented in the supplemental **Figure S8**. We revised the manuscript as following:

Result section Page 27:

*The antibiotics treatment weakened the weight loss effects of the IF treatment (**Fig. 6A**) ($p < 0.05$). The antibiotics treatment had no impact on the eWAT weight but enhanced the liver and cecum weight (**Fig. S8E-G**). Both IF and antibiotics treatment reduced food and water intake in db/db mice (**Fig. S8C, D**). However, the administration of antibiotics had no impact on the beneficial effects of IF on insulin resistance (**Fig. S8H-K**).*

Discussion section Page 37:

*Of note, antibiotics treatment alone also partly altered the food/water intake, insulin resistance, and cognitive behaviors in db/db mice (**Fig. 6, Fig. S8**), in line with a previous report revealing that metronidazole or vancomycin treatment improves brain insulin sensitivity and behavioral changes in obese and diabetic mice (Soto M, et al. Mol. Psychiatry 2018). Besides, antibiotics treatment reduced the generation of SCFAs and IPA but not the levels of TUDCA and 5-HT in IF treated db/db mice (**Fig. S9**). These results indicate*

that the efficiency of a combined treatment of an IF regimen and antibiotics in metabolic syndrome needs to be further investigated. It is also important to further investigate the molecules and pathways that transmit microbiota changes into the brain, which could lead to the discovery of new therapeutic targets for metabolism-implicated cognitive deficits.

Comment 4: The authors demonstrate that there are metabolites that associate with the improved cognitive behavior seen in fasted db/db mice and that the microbiota is required for the effect (seen with antibiotic depletion model). However, it will be important to directly test the role of these metabolites in the animal model to evaluate causation vs. correlation. The author's claim in the discussion of demonstrating "for the first time that a 28-day IF treatment alleviated diabetes-induced cognitive impairment via a microbiota-metabolites-brain axis" is not completely proven with such evidence. For example, since the authors previously identify short chain fatty acids as microbiota-associated metabolites of interest (**Fig. S4G-R**), the authors could test whether short chain fatty acids are reduced in the fasted antibiotic-treated group and if so, administer short chain fatty acids and test cognitive behavior in this group.

Response:

We acknowledge the reviewer's constructive suggestions and we certainly agree. We have now added analysis of the levels of microbiota-associated

metabolites in antibiotic-treated mice and revised the results and discussion accordingly. Specifically, our previous results showed that IF treatment increased the levels of SCFAs, IPA, and microbiota-related bile acids such as TUDCA (**Fig.4G**). Herein, we found that the removal of gut microbiota by the antibiotic treatment significantly reduced plasma IPA and fecal SCFAs in db/db mice, compared to db/db-IF group. These findings have been now added in the results and are provided in **Figure S9**.

Moreover, we performed an additional animal study to investigate the role of IF-regulated microbial metabolites, in particular, IPA and SCFAs that were remarkably reduced in the antibiotic-treated group, in improving cognitive deficits. An additional animal study was conducted to investigate the role of IF-regulated microbial metabolites, in particular, IPA and SCFAs that were reduced in the antibiotics-treated group, in improving cognitive deficits. Besides, the effects of peripheral 5-HT and TUDCA on cognitive function were also examined, since these metabolites were elevated in the IF group in current study (**Fig. 4G**). And previous studies have also reported the potential of their neuroprotective effects (Elia AE, et al., *Eur. J. Neurol.* 2016; Nunes AF et al., *Mol. Neurobiol.* 2012; Pan Q, et al. *Metab. Brain. Dis.* 2019). Administration of all these metabolites individually improved cognitive function and insulin sensitivity in db/db mice (**Fig. 6G, Fig. S10C-H**) ($p < 0.05$). Consistent with the beneficial effects of IF treatment, administration of these metabolites also enhanced mitochondrial biogenesis and protected the

ultrastructure of synapses (**Fig. 6H-J**). Moreover, treatment of TUDCA but not other tested metabolites suppressed the bodyweight gain in db/db mice (**Fig. 6F**). Besides, all the metabolites treatment had no influence on food intake and water intake of the db/db mice (**Fig. S10A, B**). Although the dosage of the metabolites treatment was higher than that in IF regimen group mice, these results partly evaluate the causation and the roles of these metabolites in during IF regimen. We revised the manuscript accordingly on **Page 28-29**.

We also added discussion on the beneficial effects of metabolites on cognitive function and mitochondrial biogenesis on **Page 36-37** as following:

Moreover, the microbial metabolites, i.e. 5-HT, TUDCA, and IPA, have been well reported to be beneficial to mitochondrial biogenesis and function. 5-HT has been reported to prevented dopamine-induced oxidative damages of mitochondria and synaptosomes (Marin IA, et al. Sci. Rep. 2017). As mentioned ahead, TUDCA is a potential neuroprotective agent (Elia AE, et al., Eur. J. Neurol. 2016; Nunes AF et al., Mol. Neurobiol. 2012). It has also been reported that TUDCA treatment prevented cognitive deficits via improving mitochondrial function and reducing neuronal apoptosis in an IPA induced model of Huntington's disease model (Nunes AF et al., Mol. Neurobiol. 2012). Similarly, as one of the microbial metabolites, IPA has been reported to protect against A β -induced neuronal death and restore mitochondrial function (Chyan Y-J, et al. J. Biol. Chem. 1999; Dragicevic N, et al. Pineal Res. 2011). SCFAs, generated from dietary fiber by gut microbiome, has been extensively studied

as potential benefits for brain health (Bourassa MW et al. Neurosci. Lett. 2016). Butyrate, for instance, was found to be a potential treatment for autism spectrum disorders as its bioactivity on enhancing mitochondrial function (Rose S, et al. Transl. Psychiat. 2018). Here, we also found treatments of microbial metabolites also improved diabetes-related cognitive deficits and increasing mitochondrial biogenesis (Fig.6).

Comment 5: Many methods are stated as being “performed as previously described”. The authors should consider adding additional methodological details in the methods section to enhance clarity.

Response:

Thanks for the comments and we agree. We have revised the Methods session in the main text. However, in order not to complicate the reading and understanding of the main text of the manuscript, we prefer to avoid presenting detailed technical methods in the manuscript, which are all well described in previous publications. Instead, we have provided the detailed description in

Supplemental materials.

Minor comments/concerns:

Comment 1: The authors should include more specific contextual information regarding prior literature in this area. For example, on pg. 4, the authors state, “They are also strongly linked to whole body metabolism and

function particularly, and have been shown to have potential neuroprotective effects”. Which metabolic effects has the gut microbiota been connected to previously and which types of neuroprotective effects?

Response:

We added more specific references for these descriptions as following:

They are also strongly linked to whole body metabolism and function particularly, and have been shown to have potential neuroprotective effects. BAs such as tauroursodeoxycholic acid (TUDCA) has been suggested as a potential agent in preventing amyotrophic lateral sclerosis and Alzheimer’s diseases (AD) (Elia AE, et al., Eur. J. Neurol. 2016; Nunes AF et al., Mol. Neurobiol. 2012). In addition, previous research demonstrated that IPA, one of the microbial deamination metabolites of tryptophan, might possess potential neuroprotective against β -amyloid-induced neuronal damage via scavenging free radicals (Chyan Y-J, et al. J. Biol. Chem. 1999).

Comment 2: Please clarify on page 18 that claudin-1 is a tight-junction protein, which is how this directly relates to intestinal permeability.

Response:

Thanks for reviewer’s suggestion. We added explanation of claudin-1 in relevant section as following on **Page 16**:

The expression of claudin-1, a tight-junction protein in gut barrier (Kinugasa T et al. Gastroenterology 2000), was also elevated in db/db-IF colon tissue (Fig. S4A, E, F), in line with the intestinal permeability alteration.

Comment 3: On page 18: “Gut microbiota was determined” specifically means “gut microbiota composition was determined...” and that this was performed by 16S rDNA sequencing. Also, it needs to be specified that while alpha diversity did not change here (Fig. 4B), beta diversity did (Figure 4C) (here the text simply states “microbiota diversity”, but this should be clarified).

Response:

Thanks for the suggestions and we have now revised description of microbiota diversity alteration to clarify the findings. Both alpha diversity (28d, i.e. at the end of the study, but not 0d, i.e. at the beginning of the study) and beta diversity were altered after IF treatment.

Revised the manuscript on **Page 16:**

Gut microbiota composition was determined from fecal samples of the respective mice on day 0 (baseline level) and day 28 using bacterial 16S rRNA gene v3-v4 amplicon sequencing. Specifically, the microbiome alpha diversity significantly increased after a 28-day IF treatment even though the total number of OTUs remained at the same level initially (Fig. 4A) ($p < 0.05$). The unweighted Unifrac distance of the db/db-IF mice on the day 28 was different from other groups, which indicates that the IF regimen changed beta diversity in the meantime (Fig. 4B) ($p < 0.05$).

Comment 4: There are many grammatical errors throughout the manuscript that require revision: page 2 "...circulated metabolites that were affect by IF", page 18, "goblet cells numbers" should be changed to "goblet cell numbers". Pg. 4: "the gut microbiota plays a vital role in inter-phasing" page 25, "A DIABLO integrative modeling was then built on the abovementioned..."

Page 31: "complications via re-structing"

Response:

Thanks for your comments. We have now corrected grammatical errors and extensively revised the manuscript.

Comment 5: Because the scale in Fig. 1N-P varies from graph to graph it is difficult to cross-compare groups. I think clarity would be enhanced here by either displaying the data separately with the same scale on the y-axis or by plotting all groups on on graph. Otherwise, it is difficult to assess the claim that the db/db-IF group "significantly decreased...the size of adipocytes" as stated on pg. 7.

Response:

Thanks and we have changed the scales in Fig.1N-P (now **Fig. S1D**) to made them consistent.

Comment 6: In Figure 1, the figure legend is inaccurate, where for example part I is listed as "fasting insulin level", but the graph is for HOMA-IR,

part J is listed as “fasting glucose”, but is actually leptin, part K is listed as HOMA-IR, but is actually 5-HT, etc.

Response:

We apologize for the mistake. As reviewer 2 suggest to simplify the figures, we revised the organization of **Figure 1** and revised the legends as well.

Comment 7: On page 10 when the following sentence is stated, “We also observed that the expression of BDNF, a neurotrophic factor that plays an essential role in maintaining neural survival and synaptic function, was increased”, this should be specified to state what the increase is relative to. A similar improvement can be made to the two sentences directly following this.

Response:

Thanks and we re-wrote these sentences as reviewer suggested as follows, on **Page 9**:

We also observed that IF treatment enhanced the expression of BDNF, a neurotrophic factor involved in maintaining neuronal survival and synaptic function, and the phosphorylation of ERK/CREB, an upstream signaling of BDNF synthesis, compared with db/db mice (**Fig. 2E**) ($p < 0.05$).

Comment 8: The title for Figure 2 can be improved to include a description of the signaling pathways that were altered with intermittent fasting.

Response:

We agree with reviewer's suggestion and the title of Figure 2 has been changed to "**Figure 2** *Intermittent fasting improved synapse ultrastructure and altered IRS/Akt and CREB/ERK signaling in db/db mice brain*".

Comment 9: In Figure 2, part I is incorrectly labelled in the legend as "G".

Response:

Thanks and we have revised this figure in the manuscript.

Reviewer #2 comments:

The study by Liu et al. describes the effects of intermittent fasting (IF) on the gut microbiome and cognitive capacity in a murine model of type 2 diabetes. There currently is a growing interest in understanding the apparent beneficial effects of fasting in relation to different disease phenotypes from a mechanistic point of view. The study of Liu et al. provides important results which further our understanding of the interrelationship between fasting, microbiome-driven metabolism and molecular effects in the brain. The study is therefore both timely and important. In general and according to my assessment, the work has been performed to a high standard. Nevertheless, I have a few reservations which may preclude publication of the manuscript in its current form.

Comment 1: Although the authors have demonstrated significant shifts in the gut microbiome, circulating metabolites and gene expression in the brain following IF and have linked these changes to improvements in cognitive function, the study fails to conclusively prove causal relationships. Granted, the administration of antibiotics and the resulting reduction in microbiome diversity and activity does provide some inroads into untangling causality but, given that the authors have identified specific microbiome-derived molecules (e.g. bile acids, serotonin, IPA, etc.) which likely confer the observed effects, I am missing the final set of experiments with these molecules to prove at least one mechanistic connection. Therefore, I would suggest that the authors add one experiment by which they would prove that fasting-induced changes to specific metabolites result in the expression changes in the brain and linked cognitive phenotypes.

Response:

We thank for reviewer's constructive suggestions and we agree. This has also been raised by Reviewer 1. As we mentioned above, we have performed analysis of the levels of microbiota-associated metabolites in antibiotic-treated mice. We have also performed additional animal studies to investigate whether metabolites, i.e. SCFAs, 3-indolepropionic acid (IPA), tauroursodeoxycholic acid (TUDCA), and serotonin (5-HT) could improve cognitive function and mitochondrial biogenesis. The results indicated that administration of these selected metabolites treatment significantly improved cognitive function in

db/db mice. In consistent with the beneficial effects of IF treatment, it has been found that the metabolites treatment also enhanced mitochondrial biogenesis and protected the ultrastructure of synapses. As mentioned ahead in the response to Reviewer 1 we added detailed description in the **Results** section (Page 28-29) and **Discussion** section (Page 36-37).

Comment 2: The figures are extremely dense (see especially Figures 1, 3 & 4) and largely illegible. I suggest the authors revise their figures and determine what information is really essential and/or which can be moved to the supplement.

Response:

Thanks for reviewer's comments and we agree. We have re-organized these figures in the main text and provided **Figure S1-S10** as supplemental figures respectively.

Comment 3: The authors discuss statistically significant differences in the text without highlighting which tests were used and without including corresponding P values. The authors should rectify this throughout the manuscript.

Response:

We appreciate reviewer's suggestion. To avoid repetitions, we prefer to provide detailed description of test methods in Statistics analysis section and

figure legends, accordingly. In the main text, we have added corresponding p values to indicate significant differences.

Comment 4& Minor comments: The text is at times difficult to follow and comprehend. It might be helpful to have it revised by a native English speaker. I make a few corresponding suggestions below.

Response:

Thanks for reviewer's suggestion. We carefully edited the language in the revised version with the help of native English speaker.

We appreciate for Editor/Reviewers' warm work earnestly. Thank you very much for your help and the manuscript has been resubmitted to your journal. We look forward to your positive response and thank you for your good comments.

Yours sincerely,

Xuebo Liu

Professor

College of Food Science and Engineering

Northwest A&F University

Tel: +86 29 87092817

Fax: +86 29 87092817

REVIEWERS' COMMENTS:

Reviewer #1 (Remarks to the Author):

Liu and colleagues analyzed the effect of intermittent fasting on cognitive behavior, glucose tolerance, hippocampal gene expression and synaptic architecture, and metabolomic. Previously, the authors tested the role of the gut microbiota in fasting-based cognitive improvements through antibiotic administration. The major critique raised by both reviewers last round was a requirement to test causation of microbiome-associated metabolites on behavior and neurophysiology. The authors demonstrate an interesting finding that 4 separate groups of microbiome-associated metabolites (TUDCA, IPA, 5-HT, and SCFAs) each improve cognition for db/db mice. While the authors did test the behavioral and neuroanatomical consequences of microbiome metabolite administration in mice, we have several comments on clarity in discussing results to improve the manuscript below.

1) For the previous comment 1 on discussing results of Figure 3A (hippocampal RNA sequencing), the authors state that while there are genes that have comparable expression between the db/db IF group and the db/m group, they prefer to highlight the genes that “increased after IF treatment compared to db/db and db/m mice”. The authors added discussion of the 483 genes in the db/db IF group that are similar to db/m and included mentioning that the KEGG analysis also demonstrates these genes are related to mitochondrial gene expression. Could the authors also please add to the discussion:

a. The percentage of these 483 genes that are similar to expression pattern to the comparison focused on by the authors (db/db IF vs. db/db and db/m).

b. Whether any of these genes/KEGG analysis categories are also similar when comparing db/m to db/db mice to isolate the disease condition. I believe this is demonstrated in the bottom of Fig 3A where 179 genes are lower for the db/db mice compared to both db/db IF and db/m and these genes include the respiratory chain and mitochondrial fission/translation. It may aid the discussion to demonstrate that when compared to db/db mice, db/m mice demonstrate improved mitochondrial metabolism and that separately, IF corrects the abnormal mitochondrial metabolism also.

2) For Figure 3B, it may be useful to either increase text size of the gene names or increase quality. Currently, this is difficult to read even when the size is increased by zooming.

3) The antibiotic paradigm for Figure 6 both prior and during fasting used includes metronidazole. About 80% of ingested metronidazole is absorbed into the plasma, and metronidazole can cross the blood brain barrier

(<https://pubchem.ncbi.nlm.nih.gov/compound/Metronidazole#section=Absorption-Distribution-and-Excretion>). The authors need to discuss therefore whether any neurobehavioral consequences

could be explained by metronidazole absorption, particularly once metronidazole crosses the blood-brain barrier.

4) In line 412: “antibiotics treatment remarkably reduced plasma IPA and fecal SCFAs”. Can you please change wording of “remarkably” to “significantly” or another neutral, statistically descriptive word? It is not surprising that these metabolites change since they are both synthesized exclusively by gut microbes.

5) Directly under the section titled “IF restructured gut microbiota and microbial metabolites”, the authors mention that endotoxemia is related to diabetes and its complications. It would be useful to mention specifically how endotoxemia relates to diabetes (i.e. increased endotoxemia from increased gut permeability can increase inflammation which triggers impaired glucose tolerance).

6) Can the authors please provide any information on how 28 days was selected for the fasting paradigm? I.e. has this timepoint previously demonstrated efficacy in mouse models on cognition, glucose tolerance, or neurophysiology?

7) In line 76, the “cognitive functions” mentioned that are affected by fasting have mainly been demonstrated in animal models and is just beginning to be studied rigorously in humans- please specify here.

Reviewer #2 (Remarks to the Author):

The authors have addressed all my previous comments in the revised version of the manuscript.

Response to Reviewers' comments

Dear Editor and Reviewers,

We sincerely appreciate for your constructive and helpful comments and suggestions to our manuscript entitled "**Gut Microbiota Mediates Intermittent-Fasting Alleviation of Diabetes-Induced Cognitive Impairment**" (NCOMMS-19-03644A). All changes are highlighted in red in the revised version of the manuscript. The corrections/revisions are listed below, point-by-point:

Reviewer #1 (Remarks to the Author):

Liu and colleagues analyzed the effect of intermittent fasting on cognitive behavior, glucose tolerance, hippocampal gene expression and synaptic architecture, and metabolomic. Previously, the authors tested the role of the gut microbiota in fasting-based cognitive improvements through antibiotic administration. The major critique raised by both reviewers last round was a requirement to test causation of microbiome-associated metabolites on behavior and neurophysiology. The authors demonstrate an interesting finding that 4 separate groups of microbiome-associated metabolites (TUDCA, IPA, 5-HT, and SCFAs) each improve cognition for db/db mice. While the authors did test the behavioral and neuroanatomical consequences of microbiome metabolite administration in mice, we have several comments on clarity in discussing results to improve the manuscript below.

Responses:

Thanks for reviewer's positive comments. We carefully responded all comments point by point as following:

Comment 1: For the previous comment 1 on discussing results of Figure 3A (hippocampal RNA sequencing), the authors state that while there are genes that have comparable expression between the db/db IF group and the db/m group, they prefer to highlight the genes that "increased after IF treatment compared to db/db and db/m mice". The authors added discussion of the 483 genes in the db/db IF group that are similar to db/m and included mentioning that the KEGG analysis also demonstrates these genes are related to mitochondrial gene expression. Could the authors also please add to the discussion:

- a. The percentage of these 483 genes that are similar to expression pattern to the comparison focused on by the authors (db/db IF vs. db/db and db/m).
- b. Whether any of these genes/KEGG analysis categories are also similar when comparing db/m to db/db mice to isolate the disease condition. I believe this is demonstrated in the bottom of Fig 3A where 179 genes are lower for the db/db mice compared to both db/db IF and db/m and these genes include the respiratory chain and mitochondrial fission/translation. It may aid the discussion to demonstrate that when compared to db/db mice, db/m mice demonstrate improved mitochondrial metabolism and that separately, IF

corrects the abnormal mitochondrial metabolism also.

Response:

a. Thanks for reviewers' constructive comments. As we described earlier, genes deposited in Group 3 and Group 6 were those significantly higher or lower expressed in both db/db mice compared to other two group, i.e. these genes had similar expressions trend in both db/m and db/db-IF group. All these 483 genes expressions were balanced by IF treatment in db/db mice.

As reviewer request, we calculated the percentage of these 483 genes that are similar to expression pattern to the comparison focused on by the authors (db/db IF vs. db/db and db/m). There were 60 genes in Group 1 were similar to Group 6, and 91 genes in Group 3 were similar to Group 4, which indicated that there were 151 genes of all 483 genes (31.3%) in db/db-IF mice were significantly upregulated or down-regulated, even compared db/m mice. We added this description in the manuscript.

b. The GO terms analysis of those genes that had lower expression in db/db mice (DEG-group 6 genes) indicated the Respiratory chain and mitochondrial fission/translation biological process were dysregulated in db/db mice, which suggested abnormal mitochondrial metabolism in diabetic mice hippocampus. However, the related mitochondria-related genes expressions were corrected by the IF regimen.

Comment 2: For Figure 3B, it may be useful to either increase text size of the

gene names or increase quality. Currently, this is difficult to read even when the size is increased by zooming.

Response:

Thanks for reviewer's comments. We have increased the font size of the text in Figure 3B.

Comment 3: The antibiotic paradigm for Figure 6 both prior and during fasting used includes metronidazole. About 80% of ingested metronidazole is absorbed into the plasma, and metronidazole can cross the blood brain barrier (<https://pubchem.ncbi.nlm.nih.gov/compound/Metronidazole#section=Absorption-Distribution-and-Excretion>). The authors need to discuss therefore whether any neurobehavioral consequences could be explained by metronidazole absorption, particularly once metronidazole crosses the blood-brain barrier.

Response:

This is a great suggestion. There are several reports indicated that metronidazole has neurotoxicity and could induce encephalopathy during the treatment. Here in our current work, metronidazole and other antibiotics were employed to treat the animals to remove the gut microbes. However, we found there were no cognitive disorders and synaptic structure damages were observed in either db/m or db/db mice (Fig. 6 B-D, Supplementary Fig. 7) after treated with antibiotics cocktails, which indicated that metronidazole and other antibiotics treatment did not triggered any neurobehavioral consequences in

present study. We added related discussion the manuscript as following:

Although there were several reports indicated that metronidazole could cross the blood-brain barrier and has potential neurotoxicity (references). However, we did not observe any behavioral disorders in both db/m and db/db mice after antibiotics treatment (Fig. 6, Supplementary Fig. 7).

References:

- 1 Tsai T-H, Chen Y-F. Pharmacokinetics of metronidazole in rat blood, brain and bile studied by microdialysis coupled to microbore liquid chromatography. J. Chromatogr. A 987, 277-282 (2003).
2. Kim DW, Park J-M, Yoon B-W, Baek MJ, Kim JE, Kim S. Metronidazole-induced encephalopathy. J. Neurol. Sci. 224, 107-111 (2004).
3. Kim J, et al. Metronidazole-induced encephalopathy in a patient with Crohn's disease. Intest. Res. 15, 124 (2017).

Comment 4: In line 412: “antibiotics treatment remarkably reduced plasma IPA and fecal SCFAs”. Can you please change wording of “remarkably” to “significantly” or another neutral, statistically descriptive word? It is not surprising that these metabolites change since they are both synthesized exclusively by gut microbes.

Response:

We revised the word “remarkably” into “significantly” in this line.

Comment 5: Directly under the section titled “IF restructured gut microbiota and microbial metabolites”, the authors mention that endotoxemia is related to diabetes and its complications. It would be useful to mention specifically how endotoxemia relates to diabetes (i.e. increased endotoxemia from increased gut permeability can increase inflammation which triggers impaired glucose tolerance).

Response:

We added the description below as reviewer suggested:

It has also been reported that increased endotoxemia from increased gut permeability can increase inflammation which triggers impaired glucose tolerance.(reference)

Reference:

4. Cani PD, et al. Changes in gut microbiota control metabolic endotoxemia-induced inflammation in high-fat diet–induced obesity and diabetes in mice. *Diabetes* 57, 1470-1481 (2008).

Comment 6: Can the authors please provide any information on how 28 days was selected for the fasting paradigm? I.e. has this timepoint previously demonstrated efficacy in mouse models on cognition, glucose tolerance, or neurophysiology?

Response:

Thanks for reviewer’s comment. We selected 4 weeks treated based on

several studies that selected 4 weeks or 30 days of alternate day fasting as intermittent fasting approach to investigate the neuroprotective effects of IF in animal models. These references were also cited in the study.

References:

5. Cignarella F, et al. Intermittent Fasting Confers Protection in CNS Autoimmunity by Altering the Gut Microbiota. *Cell Metab.* 27, 1222-1235 (2018).
6. Vasconcelos AR, et al. Intermittent fasting attenuates lipopolysaccharide-induced neuroinflammation and memory impairment. *J. Neuroinflammation* 11, 85 (2014).

Comment 7: In line 76, the “cognitive functions” mentioned that are affected by fasting have mainly been demonstrated in animal models and is just beginning to be studied rigorously in humans- please specify here.

Response:

We added the descriptions in the manuscript that these studies were done in animal models.

Reviewer #2 (Remarks to the Author):

The authors have addressed all my previous comments in the revised version of the manuscript.

Responses:

Thanks for reviewer’s positive comment.

We appreciate for Editor/Reviewers' warm work earnestly. Thank you very much for your help and the manuscript has been resubmitted to your journal. We look forward to your positive response and thank you for your good comments.

Yours sincerely,

Zhigang Liu

Associate Professor

College of Food Science and Engineering

Northwest A&F University

Tel: +86 29 87092817

Fax: +86 29 87092817